# Calibrating Translation Decoding with Quality Estimation on LLMs

**Di Wu**      **Yibin Lei**      **Christof Monz**
University of Amsterdam
`{d.wu, y.lei, c.monz}@uva.nl`

## Abstract

Neural machine translation (NMT) systems typically employ maximum *a posteriori* (MAP) decoding to select the highest-scoring translation from the distribution. However, recent evidence highlights the inadequacy of MAP decoding, often resulting in low-quality or even pathological hypotheses as the decoding objective is only weakly aligned with real-world translation quality. This paper proposes to calibrate hypothesis likelihood with translation quality from a distributional view by directly optimizing their Pearson correlation, thereby enhancing decoding effectiveness. With our method, translation with large language models (LLMs) improves substantially after limited training (2K instances per direction). This improvement is orthogonal to those achieved through supervised fine-tuning, leading to substantial gains across a broad range of metrics and human evaluations. This holds even when applied to top-performing translation-specialized LLMs fine-tuned on high-quality translation data, such as Tower, or when compared to recent preference optimization methods, like CPO. Moreover, the calibrated translation likelihood can directly serve as a strong proxy for translation quality, closely approximating or even surpassing some state-of-the-art translation quality estimation models, like CometKiwi. Lastly, our in-depth analysis demonstrates that calibration enhances the effectiveness of MAP decoding, thereby enabling greater efficiency in real-world deployment. The resulting state-of-the-art translation model, which covers 10 languages, along with the accompanying code and human evaluation data, has been released: https://github.com/moore3930/calibrating-llm-mt.

## 1 Introduction

The training of neural machine translation (NMT) has long been formulated as a maximum likelihood estimation (MLE) problem, and maximum *a posteriori* decoding (MAP) is a common decision rule, aiming to identify the highest-scoring translation. However, recent findings in NMT suggest that such a system has serious flaws. One particularly counterintuitive issue is the *beam search curse* [Koehn and Knowles, 2017, Murray and Chiang, 2018, Ott et al., 2018, Kumar and Sarawagi, 2019], where translation quality deteriorates as search approximations improve and higher-probability hypotheses are possibly worse translations.

Ott et al. [2018] explain this as a calibration issue. There is a generally low correlation between hypothesis likelihood and quality beyond a certain likelihood value. The flaws of poorly calibrated translation models can be categorized into two main aspects: First, their performance tends to be suboptimal, as better hypotheses may remain hidden within the distribution mass, inaccessible to MAP decoding [Ott et al., 2018]. Second, likelihood, as a key measure of uncertainty, can serve as a valuable indicator of translation errors within the output at test time [Wang et al., 2020, Fomicheva et al., 2020]; however, poorly calibrated models cannot effectively support this use.

Prior studies have tried to mitigate this miscalibration issue by introducing an additional optimization step during inference time, as shown in Figure 1 (top), known as quality-aware decoding (QAD) [Fer-

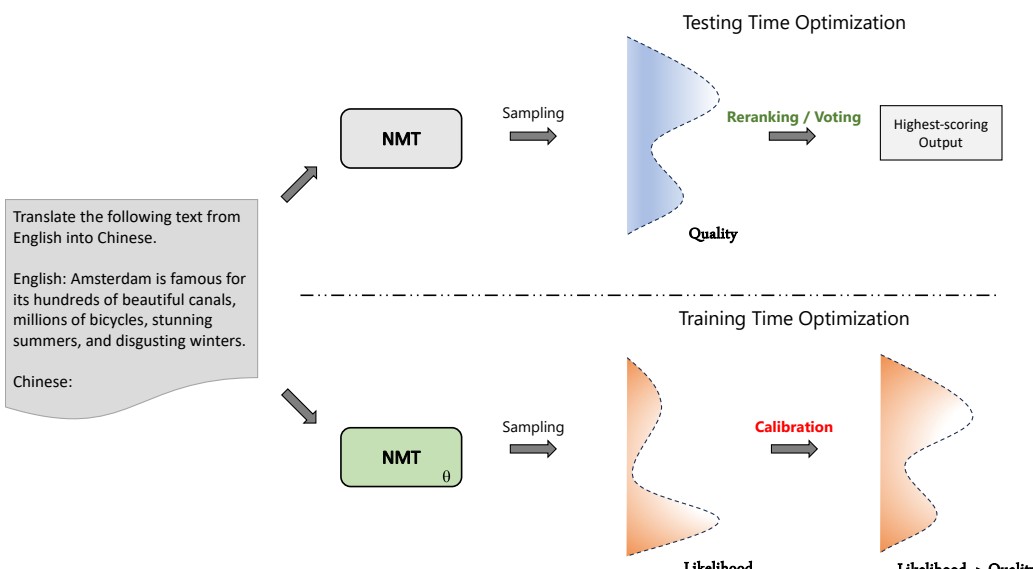

Figure 1: The illustration of quality-aware decoding (top) and our calibration method (bottom). The former explores the decoding space by adding an extra step during test-time decoding, which involves multiple rounds of sampling followed by reranking or voting to select the highest-scoring output. Our method focuses on training-time optimization, aiming to **calibrate** the likelihoods of hypotheses to their corresponding quality scores, enabling effective approximated MAP decoding.

nandes et al., 2022]. These approaches typically involve generating multiple candidate translations through sampling, followed by reranking or voting using reference-free and/or reference-based machine translation metrics, such as Best-of-N (BoN) sampling [Rei et al., 2024a, Faria et al., 2024, Brown et al., 2024, Ichihara et al., 2025] and Minimum Bayes Risk (MBR) decoding [Kumar and Byrne, 2004, Freitag et al., 2022a]. While effective, this line of methods inevitably incurs significantly higher latency. For example, the current top-performing system in the WMT24 competition [Kocmi et al., 2024a], Tower [Rei et al., 2024a], requires sampling 100 times for each prompt from a 70B LLM-based translation model, which is impractical, particularly for online environments.

This paper proposes a simple yet highly effective method to directly optimize the calibration during training, where we encourage the likelihood of translation hypotheses to align more closely with their quality by directly optimizing Pearson correlations between them. More specifically, as shown in Figure 1 (bottom), given a translation prompt $x$, we sample multiple hypotheses from an NMT model during training time and compute both their likelihoods and quality scores, as measured by any external metric, such as COMET. We then define the loss as the negative Pearson correlation between the two sets of data points and minimize it directly using a standard gradient-based optimizer. While simple, our approach yields substantial performance improvements with limited training—using only 2K instances per language—even when applied to state-of-the-art LLM-based MT systems such as Tower. Moreover, due to clear calibration improvements, the resulting models' likelihood can directly serve as a strong proxy for translation quality, even surpassing some state-of-the-art translation quality estimation (QE) models, like Comet-Kiwi [Rei et al., 2022a].

Our main contributions are summarized as follows:

- We introduce a simple yet effective method to mitigate the well-known miscalibration issue in translation during training time, where we calibrate translation decoding for quality from a holistic view by directly optimizing the Pearson correlation objective. With limited training, our models consistently outperform the strongest LLM-based MT systems, such as ALMA [Xu et al., 2023, 2024] and Tower [Rei et al., 2024a, Alves et al., 2024] series as well as the GPT-4o model, by a substantial margin across multiple automatic and human evaluation metrics. Extensive experiments demonstrate effectiveness over recent preference optimization methods, such as CPO [Xu et al., 2024], for translation.

- Our method, to some extent, unifies quality optimization and quality estimation (QE) in translation by sharing one single objective. Specifically, the uncertainty quantification of our model's output, measured by average log-likelihood, exhibits a strong correlation with human judgments, which even outperforms some state-of-the-art QE models in the community that were explicitly trained with human-annotated data, such as CometKiwi [Rei et al., 2022b]. This finding supports the view that a well-performing model should inherently "know" what constitutes a good translation.

- Our in-depth analysis demonstrates that improved likelihood-quality calibration—achieved through our calibration method—enhances the effectiveness of maximum *a posteriori* decoding, offering greater efficiency potential for real-world deployment.

## 2 Background

At a practical level, this study aims to optimize both translation quality as well as quality estimation, simultaneously in a single model. To this end, we briefly introduce the relevant foundational concepts.

**Translation metric meta-evaluation.** To automatically assess translation quality, various metrics have been proposed over the past decades, such as BLEU [Papineni et al., 2002], ChrF [Popović, 2015], COMET [Rei et al., 2020, 2022b], MetricX [Juraska et al., 2023, 2024]. Metric meta-evaluation assesses how well these metrics correlate with human judgments, where a few well-known statistics are applied, such as Pearson's $r$, Spearman's $\rho$, and Kendall's $\tau$. In this paper, we focus on calibrating hypothesis likelihood with translation quality by directly optimizing Pearson correlation, enabling simultaneous quality optimization and estimation within a single model; see §3. To ensure robust cross-metric evaluation, we report Spearman's and Kendall's scores for quality estimation; see §5.2.

**Translation quality optimization.** A few different themes investigate optimizing translation performance using signals from external metric models. We briefly summarize them as follows:

*(1) Test-time quality optimization.* To alleviate the gap between decoding objective and translation quality, Minimum Bayes Risk (MBR) decoding [Kumar and Byrne, 2004, Freitag et al., 2022a, Yang et al., 2024] suggests basing decoding decisions on statistics gathered from the distribution as a whole. As reference-free quality estimation (QE) progresses, Best-of-N sampling [Rei et al., 2024b, Faria et al., 2024, Brown et al., 2024, Ichihara et al., 2025] with a QE model becomes a straightforward strategy. However, both of these methods result in a significant amount of additional decoding time. Moreover, the risk of metric bias increases as translation improvements may stem from *metric hacking* rather than genuine quality enhancements [Skalse et al., 2022, Kovacs et al., 2024].

*(2) Preference optimization.* Recently, direct preference optimization (DPO) and its variants [Rafailov et al., 2023, Xu et al., 2024, Meng et al., 2024, Ethayarajh et al., 2024] have emerged as promising approaches for post-training LLMs, providing efficient alternatives to reinforcement learning from human feedback (RLHF). This line of methods often employs pairwise preferences under the Bradley-Terry framework [Bradley and Terry, 1952] to provide rewards, or a more general Plackett-Luce ranking framework [Plackett, 1975] when multiple ranked hypotheses are available and a few studies [Xu et al., 2024, Zhu et al., 2024] in translation also follow them. Additionally, some works [Guo et al., 2024, Lambert et al., 2024] employ these methods in an on-policy training paradigm for online LLM alignments. Notably, regardless of employing pairwise/listwise data or on-/off-policy training, the objective is generally to maximize expected rewards. A key distinction of our method, as detailed in Section 3, is that it optimizes correlation rather than expected rewards, forming the basis for unifying quality optimization and estimation. Therefore, we follow prior research [Ott et al., 2018, Zhao et al., 2022] and use the term "calibration" to highlight the difference. More detailed differences are discussed in the next section.

## 3 Translation Calibration

Formally, given a parameterized auto-regressive language model $p_\theta$ and a translation instruction $x$, the *log-likelihood* of a translation hypothesis $y_i$ is denoted as $z_\theta(y_i|x) = \log p_\theta(y_i|x)$. Meanwhile, the quality of this translation can be defined as $q(y_i|x)$ where $q$ represents any external quality evaluation model. When sampling hypotheses $y$ from $p_\theta$ conditioned on $x$, both $z_\theta(y|x)$ and $q(y|x)$ can be viewed as random variables defined over the output space. Our goal is to calibrate the likelihoods of generated hypotheses with their quality to maximize the correlation between $z_\theta(y|x)$ and $q(y|x)$.

Here, we use the statistic, i.e., *Pearson correlation coefficient* $\rho(a, b)$, to quantify the correlation. Let $a, b : \mathcal{Y} \to \mathbb{R}$ be two real-valued functions defined over a domain $\mathcal{Y}$. Their *covariance* with respect to a data distribution $p(y)$ is defined as

$$\mathrm{cov}(a, b) = \mathbb{E}_{y \sim p} \left[ (a(y) - \mathbb{E}_{y \sim p}[a(y)]) \left( b(y) - \mathbb{E}_{y \sim p}[b(y)] \right) \right]. \tag{1}$$

The corresponding Pearson score between $a$ and $b$ is given by

$$\rho(a, b) = \frac{\mathrm{cov}(a, b)}{\sqrt{\mathrm{cov}(a, a) \, \mathrm{cov}(b, b)}} = \frac{\mathbb{E}_{y \sim p} \left[ (a(y) - \mu_a) \left( b(y) - \mu_b \right) \right]}{\sigma_a \sigma_b}, \tag{2}$$

where $\mu_a$, $\mu_b$ and $\sigma_a$, $\sigma_b$ denote expectations and standard deviations, respectively. This formulation computes the correlation by normalizing the expected product of their centered values. Due to its scale-invariance and ability to capture trend consistency, the Pearson correlation coefficient is widely used in translation metric meta-evaluation.

In this study, we calculate and optimize $\rho$ with respect to the likelihood of hypotheses $z_\theta(y|x)$ and the quality score $q(y|x)$. Practically, given the intractably large decoding space, we employ Monte Carlo sampling for approximation. For each source sentence $x$, we generate $k$ hypotheses $y_i$ ($i \in \{1, \ldots, k\}$) by repeatedly prompting a large language model $\theta$ with nucleus sampling, and compute the corresponding $z_\theta(y_i|x)$ and $q(y_i|x)$, and estimate the corresponding $\mu_z, \mu_q$ and $\sigma_z, \sigma_q$.

Notably, the standard definition of correlation assumes expectations under sampling from the full distribution, i.e., unweighted correlation. Here, however, we use nucleus sampling, which introduces a biased estimate by restricting sampling to high-probability regions. This approach is motivated by two factors: (1) uniform sampling over the output space is infeasible due to its vast size, and (2) following Ott et al. [2018], we focus on correlations among likely hypotheses. Accordingly, we define the Pearson-based loss using estimates under the nucleus-induced distribution $\tilde{p}$ as follows:

$$\mathcal{L}_{\mathrm{pearson}} = -\frac{1}{k} \sum_{\substack{i=1 \\ y_i \sim \tilde{p}_\theta(\cdot|x)}}^{k} \left( \frac{z_\theta(y_i \mid x) - \mu_z}{\sigma_z} \cdot \frac{q(y_i \mid x) - \mu_q}{\sigma_q} \right). \tag{3}$$

We additionally introduce a supervised fine-tuning (SFT) term on the highest-scoring samples as a regularizer to ensure that the model's likelihood distribution remains grounded in high-quality translations, since the Pearson objective alone enforces correlation but does not constrain the absolute scale. The final loss for calibration is formulated as $\mathcal{L}_{\mathrm{cal}} = \mathcal{L}_{\mathrm{pearson}} + \mathcal{L}_{\mathrm{sft}}$.

An off-policy formulation can be obtained by trivially replacing the current model $p_\theta$ with an external model $p_{\theta*}$ for sampling. Overall, by minimizing $\mathcal{L}_{\mathrm{cal}}$, we encourage the Pearson score between $z$ and $q$ to increase. In practice, we use a gradient-based optimizer, Adam, to optimize $\theta$ for this goal, with gradients propagated through $z_\theta$, $\mu_z$, and $\sigma_z$. Despite its simplicity, several important characteristics are captured in this formulation:

- It models hypothesis qualities from a holistic view, enabling the model to make finer-grained distinctions in translation quality within the decoding space.

- It considers the value of translation quality by the metric function $q(\cdot|x)$, which is ignored in virtually all existing methods based on Bradley-Terry and Plackett-Luce, such as CPO.

- Pearson's correlation inherently applies normalization to a group of both likelihood and quality points. This normalization makes the objective invariant to scale and shift, thereby promoting stable and robust optimization across diverse input distributions.

- The objective, i.e., the Pearson's score itself, is inherently shared with that of translation metric meta-evaluation, offering a unified perspective for both quality optimization and estimation. Meanwhile, unlike other statistics like Spearman's or Kendall's scores, Pearson's coefficient is differentiable and thus suitable for gradient-based optimization frameworks.

Overall, the Pearson loss can be reduced to a mere *dot product* between two sets of normalized points, see Appendix C.1. Despite extreme simplicity, we show in the next sections that it is highly effective in simultaneously optimizing translation quality and quality estimation within a single model.

## 4 Experimental Settings

### 4.1 Base Models, Data, and Evaluation

**Base Models.** We base our experiments on two strong translation-specific LLMs, i.e., `ALMA-Base` [Xu et al., 2023] and `Tower-Base` [Alves et al., 2024], in their 7B and 13B variants. Both models achieve remarkable translation performance after post-training. For instance, the `Tower` series models [Rei et al., 2024b] achieved state-of-the-art results across all translation tasks in the WMT24 general translation tracks [Kocmi et al., 2024a], surpassing both `Google Translate` and `GPT-4`.

**Translation evaluation dataset.** For fair comparison with the `ALMA` and `Tower` model series, we evaluate translation performance on the WMT22 and WMT24 datasets [Zerva et al., 2022, Kocmi et al., 2024a], respectively. Detailed dataset descriptions can be found in Appendix D. Following current best practice [Freitag et al., 2022b, 2023], all results in this paper are measured by widely used neural metrics, involving reference-based metric COMET, and the reference-free metrics XCOMET, CometKiwi-XL, and CometKiwi-XXL[1].

**Metric meta-evaluation dataset.** To assess how well translation likelihoods correlate with human judgments, we use the sentence-level quality estimation (QE) datasets from WMT-22 [Zerva et al., 2022], where sentence pairs are annotated in multiple ways to reflect translation quality, e.g., direct assessments (DA), post-edits (PE), and multidimensional quality metrics (MQM) [Freitag et al., 2021]. In this paper, we use MQM scores as ground-truth human annotations against which the metrics' scores are evaluated, as MQM scores come from expert translators and are more reliable than the crowd-sourced DA scores. A detailed description of the dataset can be found in Appendix D.

**Calibration dataset.** For the training set, we merge all English sentences from the Flores-200 dataset [Costa-Jussà et al., 2022] in *dev* and *devtest* splits and use them as the source, consisting of 2,009 samples. For both on- and off-policy experiments, we use these sentences to construct translation prompts for each direction. The prompt templates can be found in Appendix E. For the off-policy setting, we query `gpt-4o-mini` 16 times per prompt, employing nucleus sampling with a temperature of 1.0 and a top-p of 0.98. The resulting bitexts are evaluated using varying metrics to reflect corresponding quality scores. For on-policy experiments, all settings remain the same, except that a top-k value of 5 is used for each sampling step, and the model itself is queried directly.

### 4.2 Training Setups

For all experiments, we train models using LoRA [Hu et al., 2022] with rank 8, setting $\alpha$ to 32 and dropout to 0.05. Training uses a batch size of 32, gradient accumulation of 8 steps, and sequences capped at 512 tokens. We train each model for 3 epochs, selecting checkpoints based on the best validation performance measured by XCOMET on NTREX [Federmann et al., 2022], which contains 1,997 samples per direction. To ensure robust results, we experiment with learning rates ranging from 1e-5 to 1e-4, reporting the best results for all settings. Adam [Kingma and Ba, 2014] is used as the optimizer. Unless otherwise specified (e.g., § 6.1), we use CometKiwi-XXL as signal during training and report results in XCOMET, COMET, and CometKiwi-XL. All experiments use H100 GPUs, with 7B models trained on one GPU and 13B models trained on two GPUs.

## 5 Results

### 5.1 Calibration Leads to Clear Quality Improvements

In this section, we demonstrate the effectiveness of our calibration approach by applying it to some state-of-the-art translation systems, e.g., `Tower` and `ALMA`. Table 1 presents the results for the `Tower` series under an off-policy setting (see §4.2 for the training data), measured by CometKiwi-XL, the official reference-free metric for WMT24, and XCOMET, the best performing metric as evaluated by Freitag et al. [2023]. Except for closed-source models, all results are decoded by beam search with a beam size of 5. TowerInstruct-7B/-13B, and TowerInstruct-Mistral-7B are official implementations [Rei et al., 2024a], supervised fine-tuned (SFT) on the corresponding base models using TowerBlock, a set of high-quality translation instructions constructed through careful data selection and filtering from human-annotated translation datasets, consisting of 637k instructions.

---

[1]The corresponding metric model versions are `Unbabel/wmt22-comet-da`, `Unbabel/XCOMET-XXL`, `Unbabel/wmt23-cometkiwi-da-xl`, and `Unbabel/wmt23-cometkiwi-da-xxl`, respectively.

Table 1: Evaluation of en→xx translation on WMT24 using CometKiwi-XL and XCOMET. Results are reported for all languages covered during `Tower-v1` pretraining. Note that the `Tower-v2` models, including `Tower-70B-v2`, have not been publicly released. We report their best results as published by Rei et al. [2024a]. For GPT-4o and GPT-4o-mini, we use the prompts following Hendy et al. [2023]. Results in other metrics can be found in Appendix F.1. Notably, according to Kocmi et al. [2024b], improvements of $\geq 1.99$ in XCOMET or $\geq 0.94$ in COMET scores correspond to at least 90% estimated accuracy in human judgment—both of which are achieved by our method. The best results across each model variant are bolded.

| | Models | en→de | | en→es | | en→ru | | en→zh | | en→fr | |
|---|---|---|---|---|---|---|---|---|---|---|---|
| | | KIWI-XL | XCOMET | KIWI-XL | XCOMET | KIWI-XL | XCOMET | KIWI-XL | XCOMET | KIWI-XL | XCOMET |
| *Closed* | GPT-4o-mini | 68.3 | 91.7 | 70.2 | 87.0 | 68.1 | 81.6 | 69.0 | 79.7 | 65.6 | 83.0 |
| | GPT-4o | 68.6 | 92.6 | 70.6 | 87.7 | 69.1 | 83.4 | 69.9 | 81.3 | 66.0 | 83.9 |
| | Tower-70B-v2 | – | – | – | – | – | – | – | – | – | – |
| | Tower-70B-v2 + MBR/TRR | 72.3 | – | 74.5 | – | 74.2 | – | 72.6 | – | – | – |
| | TowerInstruct-7B | 69.0 | 91.7 | 70.8 | 86.9 | 69.0 | 81.5 | 68.5 | 78.7 | 67.9 | 84.1 |
| | TowerBase-7B | – | – | – | – | – | – | – | – | – | – |
| | + SFT on BoN data | 70.0 | 92.0 | 70.8 | 86.5 | 69.6 | 81.6 | 68.4 | 77.9 | 68.0 | 83.7 |
| | + CPO | 71.1 | 93.1 | 72.0 | 87.6 | 71.6 | 83.8 | 70.4 | 80.9 | 69.3 | 85.8 |
| | + Calibration (ours) | **71.6** | **93.6** | **73.5** | **89.0** | **72.4** | **84.8** | **70.4** | **81.0** | **70.0** | **86.8** |
| | TowerInstruct-13B | 69.9 | 92.5 | 71.8 | 87.7 | 70.6 | 83.3 | 70.1 | 80.8 | 68.1 | 85.1 |
| | TowerBase-13B | – | – | – | – | – | – | – | – | – | – |
| | + SFT on BoN data | 71.1 | 92.7 | 71.8 | 87.5 | 71.3 | 82.8 | 70.1 | 80.0 | 68.0 | 84.4 |
| | + CPO | 70.5 | 92.2 | 72.0 | 87.7 | 71.9 | 84.0 | 70.3 | 81.4 | 68.8 | 85.5 |
| | + Calibration (ours) | **72.5** | **94.2** | **73.8** | **90.0** | **73.6** | **86.4** | **72.1** | **83.6** | **70.8** | **87.5** |
| | TowerInstruct-Mistral-7B | 70.0 | 92.6 | 71.9 | 87.5 | 70.3 | 83.3 | 69.6 | 80.4 | 68.3 | 84.7 |
| | + SFT on BoN data | 70.7 | 92.7 | 71.8 | 87.1 | 70.8 | 82.9 | 70.5 | 80.4 | 68.4 | 84.4 |
| | + CPO | 71.2 | 93.0 | 73.1 | 89.0 | 72.3 | 85.1 | 71.8 | 83.6 | 70.0 | 86.9 |
| | + Calibration (ours) | **72.4** | **94.0** | **73.9** | **89.9** | **73.6** | **86.1** | **72.6** | **83.7** | **70.8** | **87.4** |

| | Models | en→nl | | en→it | | en→pt | | en→ko | | Avg. | |
|---|---|---|---|---|---|---|---|---|---|---|---|
| | | KIWI-XL | XCOMET | KIWI-XL | XCOMET | KIWI-XL | XCOMET | KIWI-XL | XCOMET | KIWI-XL | XCOMET |
| *Closed* | GPT-4o-mini | 69.4 | 88.9 | 68.1 | 83.7 | 71.2 | 87.6 | 73.2 | 84.2 | 69.2 | 85.3 |
| | GPT-4o | 70.6 | 90.5 | 68.7 | 85.7 | 71.5 | 88.5 | 73.7 | 85.6 | 69.8 | 86.6 |
| | Tower-70B-v2 | – | – | – | – | – | – | – | – | – | – |
| | Tower-70B-v2 + MBR/TRR | – | – | – | – | – | – | – | – | – | – |
| | TowerInstruct-7B | 71.5 | 90.9 | 71.1 | 86.1 | 71.1 | 86.8 | 73.6 | 82.8 | 70.3 | 85.5 |
| | TowerBase-7B | – | – | – | – | – | – | – | – | – | – |
| | + SFT on BoN data | 71.5 | 89.6 | 70.8 | 85.4 | 72.5 | 87.6 | 75.7 | 84.1 | 70.8 | 85.4 |
| | + CPO | 71.9 | 90.9 | 72.2 | 86.7 | 73.4 | 88.7 | 76.1 | **87.2** | 72.0 | 87.2 |
| | + Calibration (ours) | **73.3** | **91.9** | **73.5** | **88.1** | **74.8** | **89.9** | **76.8** | **87.2** | **72.9** | **88.0** |
| | TowerInstruct-13B | 71.7 | 91.0 | 71.1 | 87.3 | 72.1 | 88.2 | 75.4 | 84.8 | 71.2 | 86.7 |
| | TowerBase-13B | – | – | – | – | – | – | – | – | – | – |
| | + SFT on BoN data | 71.7 | 90.4 | 71.6 | 86.1 | 73.0 | 88.1 | 76.2 | 85.2 | 71.6 | 86.4 |
| | + CPO | 72.3 | 90.8 | 72.5 | 87.4 | 72.2 | 86.9 | 76.9 | 87.9 | 71.9 | 87.1 |
| | + Calibration (ours) | **73.9** | **92.6** | **73.9** | **89.3** | **75.2** | **90.4** | **78.0** | **89.5** | **73.8** | **89.3** |
| | TowerInstruct-Mistral-7B | 71.9 | 91.1 | 71.6 | 87.2 | 72.1 | 88.0 | 74.2 | 85.6 | 71.1 | 86.7 |
| | + SFT on BoN data | 72.3 | 90.7 | 71.6 | 86.2 | 72.7 | 87.9 | 76.2 | 86.0 | 71.7 | 86.5 |
| | + CPO | 73.3 | 92.3 | 73.1 | 88.5 | 74.0 | 89.7 | 77.4 | 89.3 | 72.9 | 88.6 |
| | + Calibration (ours) | **74.2** | **93.2** | **74.1** | **89.6** | **75.1** | **90.7** | **78.1** | **89.7** | **73.9** | **89.4** |

We also conducted SFT on the TowerBase series using 2K Best-of-N samples per direction, selected from our calibration dataset (§4.2) based on the highest CometKiwi-XXL scores. The resulting performance is comparable to the official instruction models. When fine-tuning on the full calibration set, performance is expected to degrade, as some sampled bitext examples are of low quality.

When applying our calibration approach, very strong improvements can be observed across all directions, metrics, and base models. First, it leads to an average improvement of +2.8 points in KIWI-XL and +2.7 points in XCOMET over TowerInstruct-Mistral-7B. Additionally, Table 4 shows gains of +3.6 points in KIWI-XXL and +1.2 points in COMET, respectively. Second, this performance is comparable to that of the current top-performing system, that is Tower-70B-v2 equipped with 100-time-sampling MBR/TRR[2], while being approximately 200 times faster[3].

We also compare with CPO [Xu et al., 2024], a widely-used preference optimization method for translation, following its original setting by selecting the highest- and lowest-scoring candidates as accepted and rejected samples, respectively, achieving consistent and substantial improvements over CPO. Additionally, Appendix F.2 reports results based on the ALMA base model, showing gains of +2.0 XCOMET and +1.4 KIWI-XXL over CPO for out-of-English translation.

Note that experimental results for on-policy settings are also provided in Appendix F.3, where substantial improvements can also be consistently observed across metrics, demonstrating effectiveness under different training dynamics. Unless otherwise specified, the following analysis focuses on the off-policy setting to simplify the investigation of the properties of our calibration method.

---

[2]TRR [Rei et al., 2024a] denotes an ensemble strategy that applies reranking based on multiple metric model to select the best candidate from multiple sampled hypotheses. They report TRR results when it surpasses MBR.

[3]We roughly estimate the latency of the Tower-70B-v2 model to be 10 times that of the Tower-Mistral-7B model. Meanwhile, the former employs 100× sampling, while the latter uses beam search with a beam size of 5.

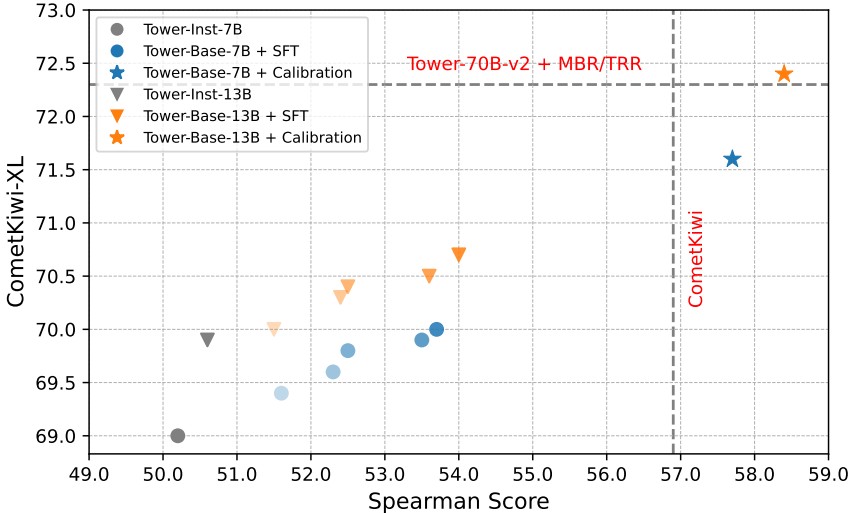

Figure 2: The Spearman coefficient and the corresponding translation performance in en→de direction under different settings for the Tower series models. The color gradients of ▼ and ●, from lighter to darker shades, indicate the results of fine-tuning with varying amounts of Best-of-N data, from 400 to 2000 samples. ★ denotes the application of our calibration method, which simultaneously surpasses both the state-of-the-art translation system and the widely used quality estimation model. Results for other languages and statistics can be found in Appendix G.

## 5.2 Calibration Leads to Direct Quality Estimation

As detailed in §3, we shared the objective for translation quality optimization and estimation, although supervisions are from machine annotations instead of human annotations. If optimized effectively, the resulting model should inherently acquire the ability to assess translation quality using hypothesis *log-likelihood* as a metric. This section evaluates how effectively calibration can elicit this capability.

We use the WMT22 metric meta-evaluation dataset [Zerva et al., 2022] and follow the official practice, see § 4.2, to assess quality estimation ability using Spearman's and Kendall's correlation. We evaluate all training directions on Tower that overlap with the WMT dataset, namely, en→de and en→ru.

Figure 2 depicts the Spearman score (metric performance) and the corresponding translation performance under different settings for Tower-7B and Tower-13B, including: (1) supervised fine-tuning using varying amounts of best-of-N samples (400/800/1200/1600/2000 samples per direction), (2) scaling the base model size from 7B to 13B, and (3) applying our calibration method. It shows that:

(1) As more Best-of-N samples are included in SFT, translation performance progressively improves. Interestingly, the quality estimation ability (Spearman scores) increases from around 51.5 to 54.0 points. We attribute this to the fact that the model assigns higher likelihoods to better hypotheses. However, these improvements are limited and not general across languages, see Appendix G.

(2) Examining the effects of scaling, we observe that: (i) scaling up from 7B to 13B generally improves translation performance for both the original TowerInstruct models and the fine-tuned models; (ii) however, its impact on calibration, i.e., quality estimation ability, remains minimal.

(3) Our calibration method manifests very strong improvements in both translation and quality estimation. For example, when applying our method to TowerBase-13B, the resulting model surpasses some state-of-the-art systems in both translation performance and quality estimation ability, i.e., Tower-70B-v2+MBR/TRR and CometKiwi, at the same time.

Results for other statistics are provided in Appendix G. Overall, we observe a clear, albeit sometimes non-linear, correlation between the models' translation performance and their quality estimation ability. These results suggest—to some extent—a unified perspective: a well-performing translation system should inherently "know" what constitutes a good translation. In turn, we also suggest optimizing translation quality by improving calibration on LLMs, rather than relying solely on extreme scaling or supervised fine-tuning, as the latter approaches show relatively limited effectiveness.

### 5.3 Calibration Enhances Effectiveness for Maximum *a Posteriori* Decoding

Compared to test-time optimization, such as Best-of-N (BoN) sampling, our approach performs sampling and directly calibrates the likelihoods of hypotheses with their translation qualities during training. Ideally, for a well-calibrated model, the potential of maximum *a posteriori* (MAP) decoding, such as beam search, should be realized more effectively.

Figure 3 compares the effectiveness of beam search (with a beam size of 5) to that of BoN sampling with varying sampling sizes, for both the baseline (`TowerInstruct-Mistral-7B`) and our calibrated models. To ensure fairness, we adopt a cross-metric evaluation for BoN sampling: Candidates are reranked using KIWI-XXL, while the best-scoring results are evaluated using COMET. Two key findings are clearly illustrated: (1) As the BoN sampling size increases, translation performance improves, with our model consistently outperforming the baseline across all settings. (2) Under the MAP decoding strategy, i.e., beam search, the baseline model (dashed blue line) underperforms compared to BoN sampling with a sampling size as small as 10. However, after calibration, beam search achieves a performance (dashed red line) comparable to that of BoN sampling with a size of 100, showing much stronger realizations of MAP capabilities. Note that beam search here is dozens of times more efficient than BoN sampling.

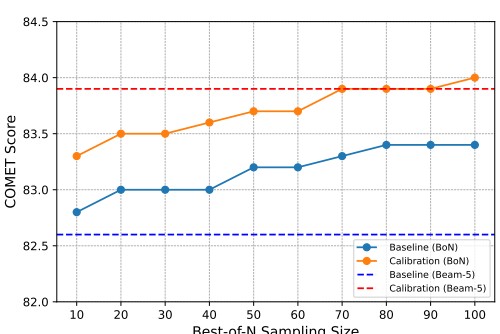

Figure 3: The average performance (en→de,es,ru,zh) measured by COMET score for both `TowerInstruct-Mistral-7B` and our calibrated model, when varying sampling size for Best-of-N (BoN) sampling and applying beam search with a beam size of 5.

Overall, compared to the state-of-the-art baseline translation system, our calibration method yields clear performance improvements and offers greater efficiency potential for real-world deployment. Unlike computation-intensive test-time optimization methods like Best-of-N sampling, it inherently improves effectiveness by calibrating decoding to the real-world quality objective, unlocking the potential of efficient MAP decoding rules, such as beam search.

## 6 Analysis

### 6.1 Cross-Metric Evaluation

Table 2: Average score differences on WMT24 over those of TowerInstruct-Mistral-7B across metrics, when applying (a) supervised fine-tuning and (b) our calibration method under different settings.

**(a) SFT Over TowerInstruct-Mistral-7B**

| Objective | KIWI-XXL | XCOMET | COMET |
|---|---|---|---|
| KIWI-XXL | +0.3 | -0.2 | -0.3 |
| XCOMET | -0.3 | +0.1 | -0.3 |
| COMET | -0.5 | -0.7 | -0.4 |

**(b) Calibration Over TowerInstruct-Mistral-7B**

| Objective | KIWI-XXL | XCOMET | COMET |
|---|---|---|---|
| KIWI-XXL | +3.5 | +2.7 | +1.2 |
| XCOMET | +2.4 | +2.8 | +1.0 |
| COMET | +2.1 | +1.4 | +0.8 |

A potential concern with directly optimizing towards translation quality is *metric gaming* [Casper et al., 2023, Kovacs et al., 2024]. To address this, we conduct experiments with TowerInstruct-Mistral-7B, using various metrics to assess hypothesis quality during training and to examine whether improvements are consistent across metrics. To maximize the contrast between metrics, we focus on three representative ones in our setting: (1) **COMET**, a strong widely used reference-based metric; (2) **KIWI-XXL**, the strongest reference-free metric in the KIWI series; and (3) the reference-free version of **XCOMET** which combines sentence-level evaluation with error span detection.

Table 2 presents the average improvements for WMT24 when applying (a) supervised fine-tuning and (b) our calibration method, using these three different metrics. It shows that: (1) When fine-tuning on our Best-of-N dataset, we generally observe slight performance degradation for all settings, except when KIWI-XXL and XCOMET are used simultaneously as both the training objective and the evaluation metric. We attribute the general degradation to the fact that TowerInstruct-Mistral-7B

is already strong, and the improvements observed in KIWI-XXL and XCOMET to the existence of slight metric gaming. (2) When applying our calibration methods, we observe consistently strong improvements across all training objectives and evaluation metrics, highlighting cross-metric effectiveness. (3) The reference-based metric, COMET, demonstrates relatively low effectiveness during training in both settings. We attribute this to the fact that reference-based metrics constrain the feasible space of positive hypotheses to be similar to the reference, thereby weakening the diversity of hypotheses recognized as valid high-quality translations.

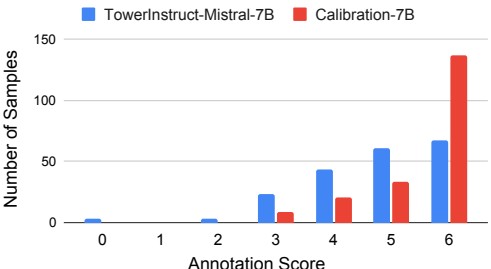

| **Human Evaluation Summary** | | | | |
| --- | --- | --- | --- | --- |
| TScore | CScore | Win | Loss | Tie |
| 4.77 | 5.49 | 106 | 32 | 62 |

Figure 4: Human evaluation results in en→zh direction: score distribution and an aggregated summary. TScore and CScore denote the average scores over 200 annotations for TowerInstruct-Mistral-7B and our calibrated model, respectively. *Win*, *Loss*, and *Tie* indicate the number of samples for which our model outperformed, underperformed, or tied with TowerInstruct-Mistral-7B.

## 6.2 Human Study

The preceding results demonstrate the effectiveness of our method across different settings and metrics. Here, we incorporate human evaluation as direct evidence for comprehensive improvements.

We randomly sample 200 translation outputs in the directions of en→zh/ru for both the baseline model (TowerInstruct-Mistral-7B) and our calibrated model from the WMT24 dataset. Several bilingual speakers, native in the respective target languages, were recruited to annotate each translation on a 0–6 scale, following the criteria outlined in Kocmi et al. [2022], see Appendix H.1.

Figure 4 shows the results for en→zh annotations, covering both fine-grained score distributions and an aggregated summary. It shows that our calibrated model yields significantly more near-perfect translations (137 vs. 67 for scores of 6), with the main gains over the baseline observed in improved handling of minor-to-major errors (denoted as score of 3/4/5) in grammar, wording, or tone, see Appendix H.2. Overall, the average score for the calibrated model is 5.49 out of 6, showing very strong system effectiveness. Compared to the state-of-the-art baseline, TowerInstruct-Mistral-7B, an overwhelming win rate (106 vs. 32) can be observed. Summaries of the annotation results for other directions are in Appendix H.3. We also release all detailed human annotations to the community[4].

## 6.3 Sensitivity to Sampling Size

As §3 mentions, we approximately characterize the decoding space through sampling. Therefore, increasing the sampling size for each source sentence is expected to improve effectiveness. To validate this, we conduct experiments on TowerInstruct-Mistral-7B, varying the sampling size during training. Also, we compared our approach with CPO and selected the highest- and lowest-scoring hypotheses as the accepted and rejected hypotheses, respectively.

Figure 5 provides the results measured by both reference-free and reference-based metrics, i.e., CometKiwi-XL and COMET, in different settings when varying the sampling size for each source sentence. It clearly shows that: (1) SFT with the best-scoring hypothesis has limited impact on translation performances; (2) CPO yields clear improvements over supervised fine-tuning, but these improvements plateau as the sample size exceeds 8; (3) our approach manifests substantial improvements over both CPO and SFT. Moreover, as the sampling size increases, continued improvements can be observed, demonstrating greater potential for further advancements. Also, the results support the underlying hypothesis that more accurate modeling of the decoding space facilitates better calibration, which in turn yields greater performance improvements.

---

[4]Detailed human annotations are available at https://huggingface.co/datasets/Calibration-Translation/Calibration-translation-human-eval.

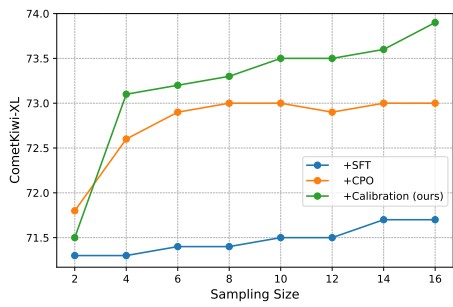
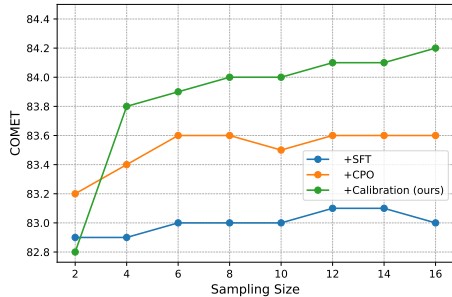

(a) KIWI-XL scores varying sampling size      (b) COMET scores varying sampling size

Figure 5: Average performance variation on the WMT24 dataset for TowerInstruct-Mistral-7B across different sampling sizes during training, measured by (a) CometKiwi-XL and (b) COMET.

## 6.4 Learning from Human Feedback

We also examine the applicability of our calibration method to human-annotated data. Specifically, we use the MAPLE dataset [Zhu et al., 2024], which includes four translation directions, as the calibration set. Each source sentence is paired with five candidate translations rated on a 1–6 scale by human experts. We compare our method against (1) SFT using the top-rated data and (2) the PL-based preference optimization approach [Zhu et al., 2024], which is designed for human preference data. Table 3 shows

Table 3: Performance comparison across SFT, PL, and calibration method on WMT22 dataset, measured by COMET.

| Model | en-de | de-en | en-zh | zh-en | Avg. |
|---|---|---|---|---|---|
| Mistral-Ins. | 81.2 | 82.7 | 82.5 | 77.7 | 81.0 |
| Mistral-Ins. + SFT | 81.8 | 83.0 | 83.3 | 78.3 | 81.6 |
| Mistral-Ins. + PL | 82.9 | 83.4 | 84.7 | 79.3 | 82.6 |
| Mistral-Ins. + Calibration | **83.5** | **84.4** | **85.4** | **80.1** | **83.4** |
| TowerMistral + SFT | 88.9 | 85.7 | 89.3 | 83.3 | 86.8 |
| TowerMistral + Calibration | **89.9** | **86.6** | **90.1** | **84.7** | **87.8** |

results for the 7B Mistral-Instruct and TowerMistral-Base models. Notably, our calibration method consistently improves over both SFT and PL baselines. Although TowerMistral-Base with SFT already substantially outperforms Mistral-Instruct, our calibration approach can further improves its performance across all directions, showing its applicability to human-annotated data.

## 7 Conclusions

This paper addresses the well-known miscalibration problem in machine translation by introducing a simple yet effective training-time method that directly optimizes the Pearson correlation objective to improve likelihood-quality calibration. Extensive experiments demonstrate several key advantages:

**Very clear and consistent performance improvements.** Our calibration method yields substantial improvements across a wide range of automatic metrics and human evaluations. For example, a calibrated Tower-7B model with beam search achieves state-of-the-art translation performance comparable to—if not exceeding—that of the much larger Tower-70B-v2 model equipped with test-time optimizations technology such as MBR decoding, all while maintaining high efficiency.

**A unified framework for quality optimization and estimation.** By using a shared objective for both tasks, our method offers a unified perspective on quality optimization and quality estimation (QE) in translation. Our empirical analysis shows a strong correlation between calibration and translation quality. Experimentally, our calibrated method achieves both top-performing translation performance and accurate quality estimation within a single model, with the latter even surpassing widely used QE models such as CometKiwi—all without relying on human-annotated data.

**Enhanced realization for maximum *a posterior* decoding.** Experimentally, we show that calibration clearly improves the effectiveness of efficient MAP decoding methods like beam search. Originally seen as less effective than Best-of-N sampling, beam search—when properly calibrated—matches the performance of Best-of-100 sampling with a beam size of just 5, significantly cutting inference cost without sacrificing quality and showing strong promise for real-world use.

## Acknowledgements

This work was funded in part by the Netherlands Organization for Scientific Research (NWO) under project number VI.C.192.080. We thank our colleagues at the University of Amsterdam, especially Sergey Troshin, Evgeniia Tokarchuk and Maya Nachesa, for their insightful discussion. D.W. thanks Chongyang for its invaluable spiritual support. The authors thank the anonymous reviewers for their constructive efforts to improve this research.

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

## A  Limitations

In this paper, we examined the impact of our calibration method under an approximate maximum *a posteriori* (MAP) decoding (i.e., beam search) across various settings. However, we did not explore decoding with the exact highest-probability output, which is computationally intractable due to the exponential search space. We also did not investigate substantially larger beam sizes (e.g., 200), which are technically feasible but incur prohibitive computational costs. We leave these directions for future work.

## B  Broader Impacts

Due to resource constraints, we focus on calibrating open-source pretrained LLMs, whose language coverage is limited to that of the base models. The impact on languages beyond this scope is left for future work. Moreover, our method inherits the limitations of the underlying models. In particular, translation quality may not be consistent across languages or demographic groups, potentially raising fairness concerns. This includes the risk of amplifying societal biases, such as gender or racial bias, that may be present in the training data.

## C  Supplementary Details on the Methodology

### C.1  Pearson Correlation Coefficient

The Pearson correlation coefficient between two sets of values $\mathbf{x}, \mathbf{y} \in \mathbb{R}^n$ is defined as:

$$\text{Pearson}(\mathbf{x}, \mathbf{y}) = \frac{\sum_{i=1}^{n}(x_i - \bar{x})(y_i - \bar{y})}{\sqrt{\sum_{i=1}^{n}(x_i - \bar{x})^2}\sqrt{\sum_{i=1}^{n}(y_i - \bar{y})^2}}, \tag{4}$$

where $\bar{x}$ and $\bar{y}$ denote the means of $\mathbf{x}$ and $\mathbf{y}$, respectively. Let us define the mean-centered and $\ell_2$-normalized versions of $\mathbf{x}$ and $\mathbf{y}$ as:

$$\tilde{\mathbf{x}} = \frac{\mathbf{x} - \bar{x}}{\|\mathbf{x} - \bar{x}\|}, \quad \tilde{\mathbf{y}} = \frac{\mathbf{y} - \bar{y}}{\|\mathbf{y} - \bar{y}\|}. \tag{5}$$

Then, the Pearson correlation simplifies to the dot product:

$$\text{Pearson}(\mathbf{x}, \mathbf{y}) = \tilde{\mathbf{x}}^{\top}\tilde{\mathbf{y}}. \tag{6}$$

This shows that the Pearson loss can be computed using only mean-centering, normalization, and a single dot product—operations that are both differentiable and computationally efficient.

## D  Detailed Dataset Description

**WMT22 and WMT24 translation datasets.** In this paper, we use the WMT22 and WMT24 datasets in the general translation track. WMT24 covers 11 language directions. In this paper, we focus on the 9 out-of-English translation directions (en→de|es|ru|zh|fr|nl|it|pt|ko) and use them to conduct experiments on Tower series models. All testsets in WMT24 are paragraph-level and share the same English parts, consisting of 960 samples for each direction. WMT22 consists of 22 language directions, covering out-, into-, and non-English translations. In this paper, for a fair comparison, we focus on 10 directions that are covered by ALMA series models, i.e., both out-of-English (en→de|cs|is|zh|ru) and into-English(en←de|cs|is|zh|ru) translations. Each direction contains 2037 sentence pairs.

**WMT22 quality estimation dataset.** To evaluate the effectiveness of using our model's likelihood as a quality estimation metric, we use the WMT 2022 Quality Estimation (QE) dataset, which provides source sentences, machine-translated outputs, and corresponding human annotations at both the sentence and word levels. The dataset includes direct assessment scores (DA), post-editing effort indicators such as HTER, word-level quality tags (OK/BAD), and multidimensional quality annotation (MQM). In this paper, we use MQM scores as the ground-truth human annotations against

which the metrics' scores are evaluated, as MQM scores come from expert translators and are more reliable than the crowdsourced DA scores. WMT22 QE dataset covers three language directions, i.e., en→de, en→ru, zh→en. We focus on the first two as they overlap with those during the Tower base model pretraining, amounting to approximately 1,000 segments per language pair. More detailed dataset descriptions can be found in Zerva et al. [2022].

# E  Prompt Templates

The prompts used in our experiments are presented in Sections E.1, E.2, and E.3. To ensure a fair comparison, the prompts strictly follow the default settings of the original models [Xu et al., 2023, Rei et al., 2024a] and prior work [Xu et al., 2024].

## E.1  GPT Prompt

> **System:**
> You are a helpful translator and only output the result.
> **User:**
> ### Translate this sentence from <source language> to <target language>, <source language>:
> <source sentence>
> ### <target language>:

## E.2  Tower Prompt

> Translate the following text from <source language> into <target language>.
> <source language>: <source sentence>
> <target language>:

## E.3  ALMA Prompt

> Translate this from <source language> into <target language>:
> <source language>: <source sentence>
> <target language>:

# F  More Results for Section 5.1

In this appendix, we provide additional results from the experiments presented in the main text, including (1) a broader range of translation evaluation metrics in Section F.1, (2) the experimental results based on the ALMA model in Section F.2, and (3) the experimental results under on-policy training dynamics in Section F.3.

## F.1  Off-Policy Results based on Tower in KIWI-XXL and COMET

Table 4 shows off-policy results measured by two other metrics, i.e., CometKiwi-XXL (abbreviated as KIWI-XXL) and COMET-22 (abbreviated as COMET). Very strong average performance improvements can be observed. For instance, +3.6 and +1.1 points of KIWI-XXL and COMET average gains are shown over TowerInstruct-Mistral-7B.

## F.2  Off-Policy Results based on ALMA

Tables 5 and 6 present the experimental results based on the ALMA model under off-policy settings. Consistently substantial improvements across metrics—COMET-22 (abbreviated as COMET), CometKiwi-XXL (abbreviated as KIWI-XXL), and XCOMET-XXL (abbreviated as XCOMET)—are

observed when applying our calibration method. The resulting performance surpasses all base-lines across nearly all language directions, including the strongest ALMA variant, ALMA-7B-R. For example, we observe gains of +0.5 COMET, +1.4 KIWI-XXL, and +2.0 XCOMET points on out-of-English translation compared to ALMA-7B-R.

### F.3 On-Policy Results based on Tower

Table 7 and Table 8 present on-policy results measured by KIWI-XL and COMET, and by KIWI-XXL and XCOMET, respectively. We found that in the on-policy setting, our calibration method performs better with a relatively smaller learning rate (1e-5), whereas in the off-policy setting, a larger learning rate (5e-5) yields the best performance. All other training settings are the same as those for off-policy, except for sampling from the model itself instead of an external model like GPT-4o-mini.

## G  More Results for Section 5.2

In this appendix, we provide additional results complementing Section 5.2 of the main text, including (1) an additional evaluation direction, en→ru, and (2) an alternative statistic for metric meta-evaluation, Kendall's $\tau$.

Figure 8 shows the Spearman coefficient and the corresponding translation performance in the en→ru direction. Meanwhile, Figures 9 and 10 present the results using Kendall's $\tau$ for the en→de and en→ru directions, respectively. It is clear that the main findings, as mentioned in Section 5.2, hold across language directions and statistics.

## H  Human Study

In this appendix, we present the detailed human evaluation results for three additional target languages: German, Russian, and Dutch. Section H.1 outlines the annotation criteria used, and Section H.3 reports the score distributions and aggregated summaries for these language directions.

### H.1 Annotation Criteria

We follow the standard of the official assessment system during WMT22 [Kocmi et al., 2022], Annotators are asked to rate a pair of source sentences and the corresponding hypothesis with scores ranging from 0 to 6. The descriptions for each quality level are as follows:

- **0: Nonsense/No meaning preserved:** Nearly all information is lost between the translation and the source. Grammar is irrelevant.
- **2: Some meaning preserved:** The translation preserves some of the meaning of the source but misses significant parts. The narrative is hard to follow due to fundamental errors. Grammar may be poor.
- **4: Most meaning preserved and few grammar mistakes:** The translation retains most of the meaning of the source. It may have some grammar mistakes or minor contextual inconsistencies.
- **6: Perfect meaning and grammar:** The meaning of the translation is completely consistent with the source and the surrounding context (if applicable). The grammar is also correct.

Note that for WMT, a continuous score is allowed for annotators. However, we require annotators to score with integers. For example, a score of 5 means in between of minor error and perfection, which may be reflected in tone, grammar, or some subtle wording.

### H.2 English-to-Chinese Translation Annotation Analysis

Figure 6 presents detailed annotation results, illustrating the changes in en→zh translations before (TowerInstruct-Mistral-7B) and after applying our calibration method. We can see that:

- Both systems perform well, as only a few translations received scores below 3.

| en-zh | | Calibrated Model | | | | | | |
|---|---|---|---|---|---|---|---|---|
| | | 0 | 1 | 2 | 3 | 4 | 5 | 6 |
| TowerInstruct-Mistral-7B | 0 | 0 | 0 | 0 | 0 | 0 | 0 | 3 |
| | 1 | 0 | 0 | 0 | 0 | 0 | 0 | 0 |
| | 2 | 0 | 0 | 0 | 1 | 1 | 0 | 1 |
| | 3 | 0 | 0 | 0 | 4 | 4 | 2 | 13 |
| | 4 | 0 | 0 | 0 | 3 | 6 | 5 | 29 |
| | 5 | 0 | 0 | 0 | 1 | 5 | 8 | 47 |
| | 6 | 0 | 0 | 0 | 0 | 5 | 18 | 44 |

Figure 6: Detailed annotation results illustrate the number of changes in en→zh translations. Each data point represents the number of samples corresponding to a specific pair of scores before and after calibration (vertical and horizontal axes), respectively.

- The primary improvements over the baseline are observed in better handling of minor to major errors. For instance, 29 and 47 translations previously rated 4 and 5 (minor errors) were upgraded to 6 (nearly perfect), respectively.

Note that annotators were blinded to the source system of each translation by **shuffling the order of each translation pair** to prevent systematic (order) bias.

## H.3   Other Annotation Results

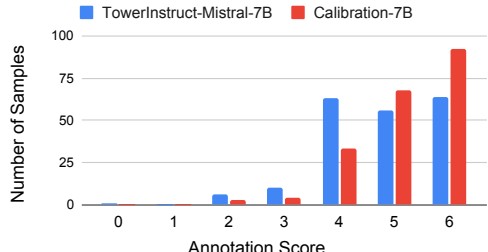

| Human Evaluation Summary | | | | |
|---|---|---|---|---|
| TScore | CScore | Win | Loss | Tie |
| 4.79 | 5.21 | 93 | 44 | 63 |

Figure 7: Human evaluation results in en→ru direction: score distribution and an aggregated summary. TScore and CScore denote the average scores over 200 annotations for TowerInstruct-Mistral-7B and our calibrated model, respectively. *Win*, *Loss*, and *Tie* indicate the number of samples for which our model outperformed, underperformed, or tied with TowerInstruct-Mistral-7B.

We will release all detailed human annotation results in other translation directions to the community[5].

---

[5]Detailed human annotations are available at `https://huggingface.co/datasets/Calibration-Translation/Calibration-translation-human-eval`.

Table 4: Evaluation of en→xx translation on WMT24 using CometKiwi-XXL and COMET. Results are reported for all languages covered during `Tower-v1` pretraining. Note that the `Tower-v2` models, including `Tower-70B-v2`, have not been publicly released. For `GPT-4o` and `GPT-4o-mini`, we use the prompts following Hendy et al. [2023]. Notably, according to Kocmi et al. [2024b], improvements of $\geq 1.99$ in XCOMET or $\geq 0.94$ in COMET scores correspond to at least 90% estimated accuracy in human judgment—both of which are achieved by our method.

| | Models | en→de | | en→es | | en→ru | | en→zh | | en→fr | |
|---|---|---|---|---|---|---|---|---|---|---|---|
| | | KIWI-XXL | COMET | KIWI-XXL | COMET | KIWI-XXL | COMET | KIWI-XXL | COMET | KIWI-XXL | COMET |
| *Closed* | GPT-4o-mini | 76.4 | 82.7 | 76.3 | 83.8 | 75.5 | 82.5 | 75.8 | 84.6 | 74.7 | 81.5 |
| | GPT-4o | 77.7 | 82.5 | 77.3 | 83.8 | 77.6 | 82.8 | 77.6 | 84.5 | 76.2 | 81.7 |
| | Tower-70B-v2 | – | – | – | – | – | – | – | – | – | – |
| | Tower-70B-v2 + MBR/TRR | – | – | – | – | – | – | – | – | – | – |
| | TowerInstruct-7B | 76.5 | 81.2 | 76.3 | 82.8 | 75.9 | 81.1 | 74.8 | 83.1 | 76.7 | 81.2 |
| | TowerBase-7B | – | – | – | – | – | – | – | – | – | – |
| | + SFT on BoN data | 77.2 | 81.3 | 75.8 | 82.4 | 76.2 | 80.9 | 74.4 | 82.4 | 76.2 | 81.0 |
| | + CPO | 78.9 | 82.2 | 78.0 | 83.2 | 78.8 | 82.2 | 77.8 | 83.4 | 78.7 | 81.2 |
| | + Calibration | **79.5** | **82.8** | **79.8** | **83.7** | **80.4** | **82.9** | **78.0** | **83.2** | **80.2** | **81.7** |
| | TowerInstruct-13B | 78.1 | 82.3 | 77.6 | 83.5 | 78.2 | 82.1 | 76.9 | 83.8 | 77.4 | 81.6 |
| | TowerBase-13B | – | – | – | – | – | – | – | – | – | – |
| | + SFT on BoN data | 79.0 | 82.3 | 77.0 | 83.1 | 78.4 | 82.0 | 76.8 | 83.8 | 77.2 | 81.5 |
| | + CPO | 79.1 | 82.1 | 78.6 | 82.5 | 80.3 | 82.6 | 78.0 | 83.4 | 79.3 | 81.5 |
| | + Calibration | **81.3** | **83.4** | **80.9** | **84.1** | **82.3** | **83.8** | **80.4** | **84.5** | **81.5** | **82.2** |
| | TowerInstruct-Mistral-7B | 78.1 | 82.0 | 77.9 | 83.0 | 77.9 | 81.8 | 76.6 | 83.8 | 77.6 | 81.5 |
| | + SFT on BoN data | 78.3 | 82.0 | 77.5 | 82.9 | 78.3 | 81.5 | 77.3 | 84.0 | 77.3 | 81.4 |
| | + CPO | 79.6 | 82.2 | 79.9 | 83.3 | 80.5 | 82.7 | 79.7 | 84.8 | 79.9 | 81.8 |
| | + Calibration | **80.7** | **83.1** | **80.6** | **83.6** | **82.0** | **83.6** | **80.4** | **84.9** | **80.8** | **82.1** |

| | Models | en→nl | | en→it | | en→pt | | en→ko | | Avg. | |
|---|---|---|---|---|---|---|---|---|---|---|---|
| | | KIWI-XXL | COMET | KIWI-XXL | COMET | KIWI-XXL | COMET | KIWI-XXL | COMET | KIWI-XXL | COMET |
| *Closed* | GPT-4o-mini | 78.3 | 84.6 | 74.1 | 83.6 | 77.9 | 81.9 | 81.2 | 86.2 | 76.7 | 83.5 |
| | GPT-4o | 80.7 | 84.6 | 76.0 | 83.8 | 79.1 | 81.9 | 82.3 | 86.2 | 78.3 | 83.5 |
| | Tower-70B-v2 | – | – | – | – | – | – | – | – | – | – |
| | Tower-70B-v2 + MBR/TRR | – | – | – | – | – | – | – | – | – | – |
| | TowerInstruct-7B | 81.1 | 84.4 | 77.7 | 83.7 | 77.9 | 81.8 | 80.0 | 84.7 | 77.4 | 82.7 |
| | TowerBase-7B | – | – | – | – | – | – | – | – | – | – |
| | + SFT on BoN data | 80.5 | 83.5 | 76.9 | 83.4 | 78.6 | 81.5 | 82.3 | 85.3 | 77.6 | 82.4 |
| | + CPO | 81.9 | 83.8 | 79.4 | 83.7 | 80.5 | 81.8 | 83.7 | 85.8 | 79.7 | 83.0 |
| | + Calibration | **83.6** | **84.8** | **81.0** | **84.3** | **81.9** | **82.7** | **84.6** | **86.1** | **81.0** | **83.6** |
| | TowerInstruct-13B | 81.4 | 84.6 | 78.4 | 84.2 | 79.1 | 82.5 | 82.9 | 85.5 | 78.9 | 83.4 |
| | TowerBase-13B | – | – | – | – | – | – | – | – | – | – |
| | + SFT on BoN data | 80.8 | 84.3 | 77.9 | 83.8 | 79.5 | 81.7 | 83.6 | 85.7 | 78.9 | 83.1 |
| | + CPO | 82.5 | 84.2 | 80.2 | 83.8 | 79.2 | 80.7 | 85.0 | 86.5 | 80.2 | 83.0 |
| | + Calibration | **84.5** | **85.1** | **82.1** | **84.6** | **82.8** | **82.7** | **86.2** | **87.1** | **82.4** | **84.2** |
| | TowerInstruct-Mistral-7B | 81.5 | 84.6 | 79.0 | 84.0 | 79.3 | 82.2 | 81.7 | 85.3 | 78.8 | 83.1 |
| | + SFT on BoN data | 81.4 | 84.2 | 78.4 | 83.7 | 79.6 | 81.7 | 83.8 | 86.1 | 79.1 | 83.0 |
| | + CPO | 83.9 | 84.8 | 80.7 | 84.0 | 81.9 | 82.2 | 85.9 | 86.9 | 81.3 | 83.6 |
| | + Calibration | **84.6** | **85.2** | **82.3** | **84.8** | **83.2** | **83.0** | **86.9** | **87.3** | **82.4** | **84.2** |

Table 5: Out-of-English translation evaluation results for ALMA models on the WMT22 dataset, measured by COMET, KIWI-XXL, and XCOMET. When applying our calibration method on ALMA-base, consistent substantial improvements across all metrics and directions can be observed.

| | en→de | | | en→cs | | |
|---|---|---|---|---|---|---|
| | COMET | KIWI-XXL | XCOMET | COMET | KIWI-XXL | XCOMET |
| ALMA-7B-LoRA | 85.45 | 80.70 | 96.49 | 89.05 | 82.06 | 90.82 |
| + SFT on preferred data | 85.42 | 80.44 | 96.26 | 89.11 | 81.28 | 90.26 |
| + DPO | 85.19 | 80.02 | 96.22 | 88.78 | 81.03 | 90.12 |
| + CPO (ALMA-7B-R) | 86.06 | 82.77 | 97.11 | 89.61 | 84.81 | 91.91 |
| + Calibration (ours) | **86.22** | **83.83** | **97.25** | **90.24** | **86.25** | **93.06** |

| | en→is | | | en→zh | | |
|---|---|---|---|---|---|---|
| | COMET | KIWI-XXL | XCOMET | COMET | KIWI-XXL | XCOMET |
| ALMA-7B-LoRA | 85.44 | 81.51 | 89.94 | 84.87 | 77.14 | 88.11 |
| + SFT on preferred data | 85.19 | 80.25 | 89.15 | 85.36 | 78.16 | 88.34 |
| + DPO | 85.20 | 80.42 | 88.97 | 84.73 | 76.96 | 87.72 |
| + CPO (ALMA-7B-R) | 85.80 | 82.35 | 89.63 | 85.89 | 81.79 | 89.55 |
| + Calibration (ours) | **86.29** | **84.11** | **91.62** | **86.47** | **83.18** | **90.75** |

| | en→ru | | | Avg. | | |
|---|---|---|---|---|---|---|
| | COMET | KIWI-XXL | XCOMET | COMET | KIWI-XXL | XCOMET |
| ALMA-7B-LoRA | 87.05 | 82.60 | 92.98 | 86.37 | 80.80 | 91.67 |
| + SFT on preferred data | 86.88 | 81.79 | 92.57 | 86.39 | 80.38 | 91.32 |
| + DPO | 86.70 | 81.61 | 92.53 | 86.12 | 80.01 | 91.11 |
| + CPO (ALMA-7B-R) | 87.86 | 84.97 | 94.15 | 87.04 | 83.34 | 92.47 |
| + Calibration (ours) | **88.41** | **86.30** | **94.46** | **87.53** | **84.73** | **94.43** |

Table 6: Into-English translation evaluation results for ALMA models on the WMT22 dataset, measured by COMET, KIWI-XXL, and XCOMET. When applying our calibration method on ALMA-base, consistent substantial improvements across all metrics and directions can be observed.

| | de→en | | | cs→en | | |
|---|---|---|---|---|---|---|
| | COMET | KIWI-XXL | XCOMET | COMET | KIWI-XXL | XCOMET |
| ALMA-7B-LoRA | 83.95 | 82.58 | 92.35 | 85.93 | 81.42 | 81.34 |
| + SFT on preferred data | 84.39 | 82.72 | 93.19 | 86.17 | 81.95 | 84.55 |
| + DPO | 84.02 | 82.47 | 92.26 | 85.87 | 81.30 | 81.10 |
| + CPO (ALMA-7B-R) | 84.61 | 83.11 | 93.85 | 86.29 | 82.29 | **85.76** |
| + Calibration (ours) | **85.03** | **84.30** | **94.33** | **86.52** | **84.13** | 84.79 |

| | is→en | | | zh→en | | |
|---|---|---|---|---|---|---|
| | COMET | KIWI-XXL | XCOMET | COMET | KIWI-XXL | XCOMET |
| ALMA-7B-LoRA | 86.09 | 84.65 | 75.02 | 79.78 | 73.65 | 83.94 |
| + SFT on preferred data | 86.47 | 85.23 | 78.87 | 80.50 | 74.91 | 89.81 |
| + DPO | 85.96 | 84.44 | 75.19 | 79.91 | 73.51 | 89.22 |
| + CPO (ALMA-7B-R) | 86.66 | 85.13 | **79.14** | 80.95 | 75.72 | 90.74 |
| + Calibration (ours) | **87.29** | **86.37** | 79.07 | **81.33** | **76.88** | **92.61** |

| | ru→en | | | Avg. | | |
|---|---|---|---|---|---|---|
| | COMET | KIWI-XXL | XCOMET | COMET-22 | KIWI-XXL | XCOMET |
| ALMA-7B-LoRA | 84.84 | 80.19 | 88.50 | 84.12 | 80.50 | 84.23 |
| + SFT on preferred data | 85.00 | 80.47 | 89.54 | 84.51 | 81.06 | 87.19 |
| + DPO | 84.71 | 80.04 | 88.34 | 84.09 | 80.35 | 85.22 |
| + CPO (ALMA-7B-R) | 85.11 | 80.69 | 90.10 | 84.72 | 81.39 | 87.92 |
| + Calibration (ours) | **85.12** | **81.87** | **90.56** | **85.06** | **82.71** | **88.27** |

Table 7: On-policy results of en→xx translation on WMT24 using CometKiwi-XL and XCOMET. Results are reported for all languages covered during Tower-v1 pretraining.

| | | en→de | | en→es | | en→ru | | en→zh | | en→fr | |
|---|---|---|---|---|---|---|---|---|---|---|---|
| | Models | KIWI-XL | XCOMET | KIWI-XL | XCOMET | KIWI-XL | XCOMET | KIWI-XL | XCOMET | KIWI-XL | XCOMET |
| *Closed* | GPT-4o-mini | 68.3 | 91.7 | 70.2 | 87.0 | 68.1 | 81.6 | 69.0 | 79.7 | 65.6 | 83.0 |
| | GPT-4o | 68.6 | 92.6 | 70.6 | 87.7 | 69.1 | 83.4 | 69.9 | 81.3 | 66.0 | 83.9 |
| | Tower-70B-v2 | – | – | – | – | – | – | – | – | – | – |
| | Tower-70B-v2 + MBR/TRR | 72.3 | – | 74.5 | – | 74.2 | – | 72.6 | – | – | – |
| | TowerInstruct-7B | 69.0 | 91.7 | 70.8 | 86.9 | 69.0 | 81.5 | 68.5 | 78.7 | 67.9 | 84.1 |
| | TowerBase-7B | – | – | – | – | – | – | – | – | – | – |
| | + SFT on BoN data | 69.8 | 92.0 | 71.6 | 87.2 | 69.2 | 81.8 | 68.6 | 78.6 | 68.7 | 84.0 |
| | + SFT on CPO | 70.2 | 92.1 | 72.5 | 89.0 | 71.8 | 84.6 | 70.0 | 80.4 | 69.4 | 86.3 |
| | + Calibration | **71.6** | **93.6** | **74.0** | **90.1** | **73.1** | **85.7** | **71.6** | **81.8** | **70.9** | **87.5** |
| | TowerInstruct-13B | 69.9 | 92.5 | 71.8 | 87.7 | 70.6 | 83.3 | 70.1 | 80.8 | 68.1 | 85.1 |
| | TowerBase-13B | – | – | – | – | – | – | – | – | – | – |
| | + SFT on BoN data | 70.6 | 92.9 | 71.9 | 88.8 | 71.2 | 84.3 | 70.3 | 81.6 | 68.3 | 86.0 |
| | + CPO | 70.5 | 92.4 | 73.2 | 88.4 | 72.0 | 84.4 | 70.6 | 82.1 | 68.8 | 85.9 |
| | + Calibration | **71.5** | **94.2** | **74.3** | **90.3** | **73.3** | **86.2** | **72.0** | **83.9** | **70.2** | **87.3** |
| | TowerInstruct-Mistral-7B | 70.0 | 92.6 | 71.9 | 87.5 | 70.3 | 83.3 | 69.6 | 80.4 | 68.3 | 84.7 |
| | + SFT on BoN data | 70.4 | 92.8 | 72.5 | 88.4 | 71.0 | 84.3 | 70.7 | 81.2 | 69.0 | 85.6 |
| | + CPO | 70.7 | 92.4 | 72.4 | 88.0 | 72.1 | 84.2 | 71.7 | 82.5 | 69.3 | 85.6 |
| | + Calibration | **71.8** | **93.9** | **74.1** | **90.0** | **73.5** | **86.3** | **72.5** | **84.0** | **70.5** | **87.0** |

| | | en→nl | | en→it | | en→pt | | en→ko | | Avg. | |
|---|---|---|---|---|---|---|---|---|---|---|---|
| | Models | KIWI-XL | XCOMET | KIWI-XL | XCOMET | KIWI-XL | XCOMET | KIWI-XL | XCOMET | KIWI-XL | XCOMET |
| *Closed* | GPT-4o-mini | 69.4 | 88.9 | 68.1 | 83.7 | 71.2 | 87.6 | 73.2 | 84.2 | 69.2 | 85.3 |
| | GPT-4o | 70.6 | 90.5 | 68.7 | 85.7 | 71.5 | 88.5 | 73.7 | 85.6 | 69.8 | 86.6 |
| | Tower-70B-v2 | – | – | – | – | – | – | – | – | – | – |
| | Tower-70B-v2 + MBR/TRR | – | – | – | – | – | – | – | – | – | – |
| | TowerInstruct-7B | 71.5 | 90.9 | 71.1 | 86.1 | 71.1 | 86.8 | 73.6 | 82.8 | 70.3 | 85.5 |
| | TowerBase-7B | – | – | – | – | – | – | – | – | – | – |
| | + SFT on BoN data | 71.8 | 90.9 | 71.5 | 86.3 | 71.2 | 87.0 | 74.5 | 82.8 | 70.8 | 85.6 |
| | + CPO | 72.2 | 91.2 | 72.5 | 87.7 | 73.7 | 88.9 | 75.3 | 86.4 | 71.8 | 87.4 |
| | + Calibration | **73.7** | **92.3** | **73.9** | **88.7** | **75.0** | **90.5** | **76.7** | **87.5** | **73.4** | **88.6** |
| | TowerInstruct-13B | 71.7 | 91.0 | 71.1 | 87.3 | 72.1 | 88.2 | 75.4 | 84.8 | 71.2 | 86.7 |
| | TowerBase-13B | – | – | – | – | – | – | – | – | – | – |
| | + SFT on BoN data | 72.0 | 91.6 | 71.2 | 87.8 | 72.6 | 88.6 | 75.6 | 86.8 | 71.5 | 87.6 |
| | + CPO | 72.1 | 91.6 | 72.7 | 87.9 | 73.4 | 88.6 | 76.4 | 87.5 | 72.2 | 87.6 |
| | + Calibration | **73.5** | **93.2** | **74.2** | **89.5** | **74.4** | **90.1** | **77.4** | **89.4** | **73.4** | **89.3** |
| | TowerInstruct-Mistral-7B | 71.9 | 91.1 | 71.6 | 87.2 | 72.1 | 88.0 | 74.2 | 85.6 | 71.1 | 86.7 |
| | + SFT on BoN data | 72.5 | 91.7 | 72.3 | 87.7 | 72.8 | 88.6 | 75.5 | 86.9 | 71.8 | 87.5 |
| | + CPO | 73.4 | 92.3 | 73.0 | 87.8 | 72.9 | 88.3 | 77.4 | 88.3 | 72.5 | 87.7 |
| | + Calibration | **73.9** | **93.2** | **74.0** | **89.3** | **74.7** | **90.2** | **77.7** | **89.4** | **73.6** | **89.3** |

Table 8: On-policy results of `en→xx` translation on WMT24 using CometKiwi-XXL and COMET. Results are reported for all languages covered during `Tower-v1` pretraining.

| Models | en→de KIWI-XXL | en→de COMET | en→es KIWI-XXL | en→es COMET | en→ru KIWI-XXL | en→ru COMET | en→zh KIWI-XXL | en→zh COMET | en→fr KIWI-XXL | en→fr COMET |
|---|---|---|---|---|---|---|---|---|---|---|
| *Closed* GPT-4o-mini | 76.4 | 82.7 | 76.3 | 83.8 | 75.5 | 82.5 | 75.8 | 84.6 | 74.7 | 81.5 |
| GPT-4o | 77.7 | 82.5 | 77.3 | 83.8 | 77.6 | 82.8 | 77.6 | 84.5 | 76.2 | 81.7 |
| Tower-70B-v2 | – | – | – | – | – | – | – | – | – | – |
| Tower-70B-v2 + MBR/TRR | – | – | – | – | – | – | – | – | – | – |
| TowerInstruct-7B | 76.5 | 81.2 | 76.3 | 82.8 | 75.9 | 81.1 | 74.8 | 83.1 | 76.7 | 81.2 |
| TowerBase-7B | – | – | – | – | – | – | – | – | – | – |
|   + SFT on BoN data | 76.7 | 81.4 | 76.6 | 82.4 | 76.1 | 81.7 | 75.4 | 82.2 | 77.3 | 81.2 |
|   + CPO | 78.3 | 81.1 | 79.7 | 82.7 | 80.0 | 82.1 | 77.8 | 82.9 | 79.8 | 80.9 |
|   + Calibration | **79.8** | **82.2** | **80.8** | **83.4** | **81.2** | **83.2** | **79.4** | **83.8** | **81.4** | **81.8** |
| TowerInstruct-13B | 78.1 | 82.3 | 77.6 | 83.5 | 78.2 | 82.1 | 76.9 | 83.8 | 77.4 | 81.6 |
| TowerBase-13B | – | – | – | – | – | – | – | – | – | – |
|   + SFT on BoN data | 78.7 | 81.5 | 78.5 | 83.4 | 79.6 | 82.2 | 78.4 | 83.4 | 78.7 | 81.6 |
|   + CPO | 79.4 | 82.7 | 79.7 | 82.5 | 80.9 | 82.8 | 79.2 | 83.2 | 79.7 | 81.0 |
|   + Calibration | **81.0** | **83.2** | **81.4** | **83.4** | **82.4** | **83.5** | **80.7** | **84.3** | **81.3** | **81.7** |
| TowerInstruct-Mistral-7B | 78.1 | 82.0 | 77.9 | 83.0 | 77.9 | 81.8 | 76.6 | 83.8 | 77.6 | 81.5 |
|   + SFT on BoN data | 78.6 | 82.5 | 78.6 | 83.1 | 79.0 | 82.4 | 77.9 | 84.3 | 78.6 | 81.9 |
|   + CPO | 78.7 | 81.9 | 79.1 | 82.2 | 80.3 | 82.5 | 79.4 | 84.3 | 79.7 | 81.3 |
|   + Calibration | **80.7** | **83.0** | **81.3** | **83.6** | **82.0** | **83.6** | **80.6** | **84.6** | **81.2** | **81.8** |

| Models | en→nl KIWI-XXL | en→nl COMET | en→it KIWI-XXL | en→it COMET | en→pt KIWI-XXL | en→pt COMET | en→ko KIWI-XXL | en→ko COMET | Avg. KIWI-XXL | Avg. COMET |
|---|---|---|---|---|---|---|---|---|---|---|
| *Closed* GPT-4o-mini | 78.3 | 84.6 | 74.1 | 83.6 | 77.9 | 81.9 | 81.2 | 86.2 | 76.7 | 83.5 |
| GPT-4o | 80.7 | 84.6 | 76.0 | 83.8 | 79.1 | 81.9 | 82.3 | 86.2 | 78.3 | 83.5 |
| Tower-70B-v2 | – | – | – | – | – | – | – | – | – | – |
| Tower-70B-v2 + MBR/TRR | – | – | – | – | – | – | – | – | – | – |
| TowerInstruct-7B | 81.1 | 84.4 | 77.7 | 83.7 | 77.9 | 81.8 | 80.0 | 84.7 | 77.4 | 82.7 |
| TowerBase-7B | – | – | – | – | – | – | – | – | – | – |
|   + SFT on BoN data | 81.4 | 84.9 | 78.0 | 84.5 | 78.7 | 82.5 | 80.2 | 85.0 | 77.8 | 82.9 |
|   + CPO | 82.6 | 83.9 | 80.9 | 83.8 | 81.2 | 82.3 | 83.3 | 85.5 | 80.4 | 82.8 |
|   + Calibration | **84.1** | **84.7** | **82.1** | **84.9** | **82.9** | **83.5** | **84.4** | **86.2** | **81.8** | **83.7** |
| TowerInstruct-13B | 81.4 | 84.6 | 78.4 | 84.2 | 79.1 | 82.5 | 82.9 | 85.5 | 78.9 | 83.4 |
| TowerBase-13B | – | – | – | – | – | – | – | – | – | – |
|   + SFT on BoN data | 82.2 | 84.4 | 80.2 | 85.1 | 80.4 | 83.3 | 83.9 | 85.8 | 80.1 | 83.4 |
|   + CPO | 83.2 | 83.7 | 80.5 | 84.1 | 81.4 | 82.3 | 85.1 | 86.5 | 81.0 | 83.2 |
|   + Calibration | **84.8** | **84.6** | **82.2** | **84.6** | **83.0** | **83.0** | **86.5** | **87.1** | **82.6** | **83.9** |
| TowerInstruct-Mistral-7B | 81.5 | 84.6 | 79.0 | 84.0 | 79.3 | 82.2 | 81.7 | 85.3 | 78.8 | 83.1 |
|   + SFT on BoN data | 82.2 | 84.7 | 79.8 | 84.6 | 80.2 | 82.6 | 83.3 | 86.3 | 79.8 | 83.6 |
|   + CPO | 83.6 | 84.6 | 80.8 | 84.0 | 80.8 | 82.3 | 85.6 | 86.7 | 80.9 | 83.3 |
|   + Calibration | **84.8** | **84.9** | **82.1** | **84.7** | **82.7** | **83.1** | **86.1** | **87.2** | **82.4** | **84.1** |

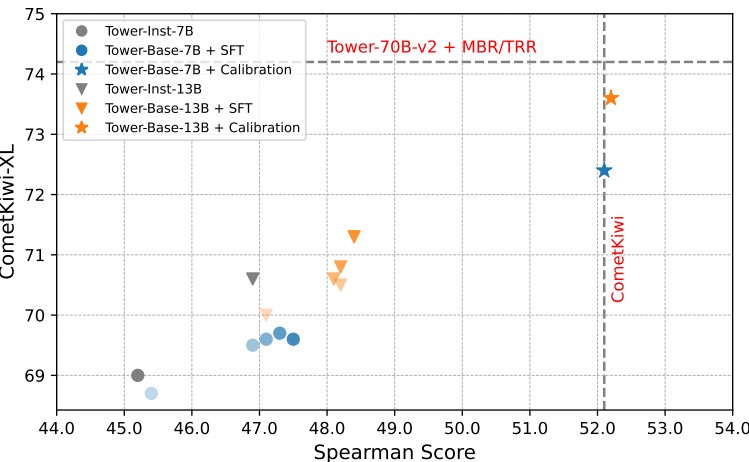

Figure 8: The Spearman coefficient and the corresponding translation performance in `en→ru` direction under different settings for the Tower series models. The color gradients of ▼ and ●, from lighter to darker shades, indicate the results of fine-tuning with varying amounts of Best-of-N data, from 400 to 2000 samples. ★ and ★ denote the application of our calibration method on 13B and 7B models, respectively.

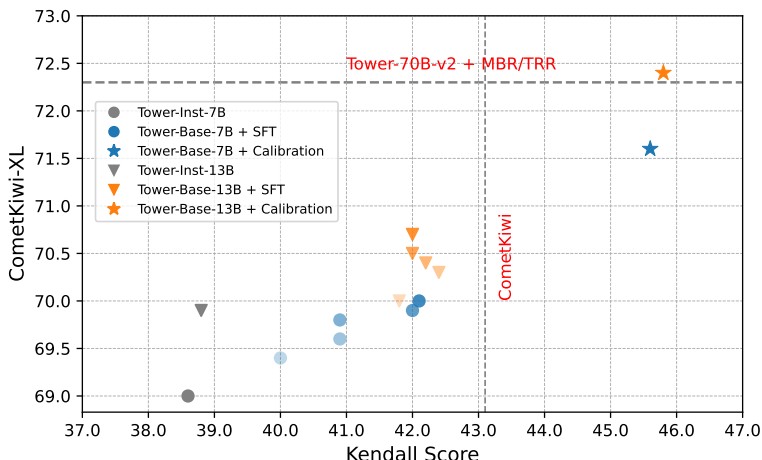

Figure 9: The Kendall coefficient and the corresponding translation performance in en→de direction under different settings for the Tower series models. The color gradients of ▼ and ●, from lighter to darker shades, indicate the results of fine-tuning with varying amounts of Best-of-N data, from 400 to 2000 samples. ★ and ★ denote the application of our calibration method on 13B and 7B models, respectively.

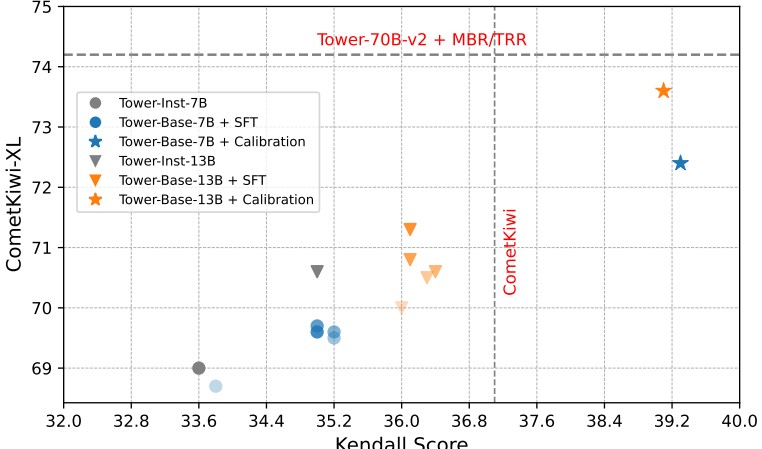

Figure 10: The Kendall coefficient and the corresponding translation performance in en→ru direction under different settings for the Tower series models. The color gradients of ▼ and ●, from lighter to darker shades, indicate the results of fine-tuning with varying amounts of Best-of-N data, from 400 to 2000 samples. ★ and ★ denote the application of our calibration method on 13B and 7B models, respectively.

