# OpenReview forum: "Calibrating Translation Decoding with Quality Estimation on LLMs"
_NeurIPS.cc/2025/Conference — NeurIPS 2025 poster_

### Official Review · Reviewer_RNS9 · 2025-06-17

**Clarity:** 3
**Significance:** 3
**Originality:** 3
**Rating:** 5
**Confidence:** 4

**Summary:**

The presented paper proposes an additional loss for machine translation fine-tuning of LLMs which directly optimizes the spearman correlation towards neural translation quality metrics. The authors demonstrate that this approach leads to a better calibrated model with superior translation quality that can also serve as a quality estimator based on its hypothesis likelihoods.

**Questions:**

N/A

**Ethical Concerns:**

["NO or VERY MINOR ethics concerns only"]

**Final Justification:**

All my concerns have been addressed in the rebuttal specifically with additional experiments around PL, paragraph level results, and clarifications on the comparison.

**Limitations:**

Yes

**Quality:**

2

**Strengths And Weaknesses:**

### Strengths

- Interesting finding that we can directly optimize Pearson correlation to not only get a better translation model but also a better QE metric.
- Following best practices in machine translation e.g. multiple neural metrics and the experiments are comprehensive.
- Interesting finding that scaling up the calibrated model improves translation quality but not quality estimation


### Weaknesses

- l. 150 the statement that the translation quality has not been used in previous works is not exactly accurate, see for example "[A Preference-driven Paradigm for Enhanced Translation with Large Language Models](https://aclanthology.org/2024.naacl-long.186/)" (Zhu et al., NAACL 2024).
- If I understand correctly, the proposed approach uses listwise preference data through the proxy of the $L_{pearson}$ which is not really a fair comparison to e.g. CPO which only leverages a single rejected hypothesis. A more fair comparison would be towards "[A Preference-driven Paradigm for Enhanced Translation with Large Language Models](https://aclanthology.org/2024.naacl-long.186/)" (Zhu et al., NAACL 2024) where we leverage the metric scores as replacement for human annotations and optimize directly on listwise rejected hypothesis.
- The presented approach is only evaluated on the sentence level and doesn't venture into paragraph or document level machine translation which may come with different learning dynamics.
- l. 218ff. / l.333f.: I don't think it is justified to say that the quality is "comparable to that of the current top-performing system, that is Tower-70B-v2 equipped with 100-time-sampling MBR/TRR"as there are still significant gaps towards that from Table 1 which are as big (and sometimes even bigger) than the gain achieved through this approach compared to CPO.

---

> ### Author Rebuttal · Authors · 2025-07-31
>
> Thanks for your careful reading and insightful suggestions, and for noting our contribution. We summarize and address your concerns as follows:
>
> > "l. 150 the statement that the translation quality has not been used in previous works is not exactly accurate, see for example "A Preference-driven Paradigm for Enhanced Translation with Large Language Models" (Zhu et al., NAACL 2024)."
>
> We agree and appreciate the clarification. We had misread the reference, thinking it referred to the original PL model. In fact, they reweight each pairwise comparison by the score difference. Thank you for pointing this out. We will revise our wording accordingly and include a discussion in the background section.
>
> > "If I understand correctly, the proposed approach uses listwise preference data through the proxy of the Pearson Loss, which is not really a fair comparison to e.g. CPO which only leverages a single rejected hypothesis. A fairer comparison would be towards A Preference-driven Paradigm for Enhanced Translation with Large Language Models (Zhu et al., NAACL 2024) where we leverage the metric scores as replacement for human annotations and optimize directly on listwise rejected hypothesis."
>
> Thank you for your suggestion. We agree that a direct comparison would ideally strengthen our work. Unfortunately, [Zhu et al., 2024] did not release their codebase, and their implementation is based on weaker models such as LLaMA-1 or Mistral. To address your concern, we apply our method on Mistral using their data (WMT data and human annotations). Also, we report our results when applying calibration to the top-performing translation model, i.e., TowerMistral-7B [1], using their data.
>
> To ensure comparability, we use the same beam search size of 4. The results are evaluated on the WMT22 test set using COMET as shown in the following table:
>
> |                            | EN-DE | DE-EN | EN-ZH | ZH-EN | Avg. |
> |----------------------------|:-----:|:-----:|:-----:|:-----:|:----:|
> | Mistral-Ins.               | 81.2  | 82.7  | 82.5  | 77.7  | 81.0 |
> | Mistral-Ins. + BEST        | 81.8  | 83.0  | 83.3  | 78.3  | 81.6 |
> | Mistral-Ins. + PL          | 82.9  | 83.4  | 84.7  | 79.3  | 82.6 |
> | **Mistral-Ins. + Calibration** | **83.5** | **84.4** | **85.4** | **80.1** | **83.4** |
> | TowerMistral + BEST        | 88.9  | 85.7  | 89.3  | 83.3  | 86.8 |
> | **TowerMistral + Calibration** | **89.9** | **86.6** | **90.1** | **84.7** | **87.8** |
>
> \+ Best is supervised fine-tuning using best scoring samples, and \+ PL is the result of their implementation of the PL model. \+ Calibration is applying our method. It is easy to see that
>
> 1. On the weaker base models, such as Mistral-Ins, **Calibration yields clear improvements** over both SFT and PL. We **provisionally attribute** this gain to Calibration’s broader consideration of factors compared to PL. Specifically: (1) Calibration normalizes both likelihood and quality scores within a group, making the objective invariant to scale and shift. This promotes stable and robust optimization across diverse input distributions. (2) Calibration directly integrates translation quality via the metric function q(⋅∣x), and optimizes at the distribution level using the Pearson correlation. In contrast, [Zhu et al.] introduce reweighting for each pairwise comparison in PL modeling. In extreme cases, this approach may treat a comparison between ratings 2 (bad) and 1 (nonsense) the same as one between 4 (nearly perfect) and 3 (good), which might be overly heuristic.
> 2. Even when applying calibration to the top-performing system, TowerMistral, using human-annotated data, our system still achieves clear, albeit relatively smaller, gains, demonstrating that calibration training is also feasible with human annotations rather than machine-generated signals.
>
> > "The presented approach is only evaluated on the sentence level and doesn't venture into paragraph or document level machine translation which may come with different learning dynamics."
>
> To address your concern, we collect paragraph-level data in four high-resource language pairs (en-de, en-fr, en-zh, and en-ru) from WMT24++ [1] for evaluation, following current representative practices for long-form data [2]. We merge sentence-level segments into paragraphs using the WMT24++ metadata. This process yields 221 paragraphs, each capped at 150 space-separated tokens in English to ensure compatibility with current evaluation metrics [3]. We follow all settings the same as those in Table 1 based on TowerMistral-7B. We report the results in three metrics, i.e., KIWI-XL, KIWI-XXL, and COMET, as follows:
>
> |             | en-de | en-fr | en-zh | en-ru | Avg. |
> |-------------|:-----:|:-----:|:-----:|:-----:|:----:|
> | SFT (KIWI-XL)    |  68.3 |  65.5 |  69.0 |  66.2 | 67.2 |
> | CPO (KIWI-XL)        |  67.9 |  66.1 |  69.8 |  66.6 | 67.6 |
> | **Calibration (KIWI-XL)** |  **69.4** |  **66.6** |  **70.2** |  **68.4** | **68.6** |
> |             |       |       |       |       |      |
> | SFT (KIWI-XXL)   |  77.9 |  77.4 |  76.2 |  75.1 | 76.6 |
> | CPO (KIWI-XXL)         |  77.0 |  78.6 |  77.6 |  76.0 | 77.3 |
> | **Calibration (KIWI-XXL)** |  **79.3** | **79.2** | **78.2** | **78.0** | **78.6** |
> |             |       |       |       |       |      |
> | SFT (COMET) |  80.1 |  79.5 |  82.3 |  80.7 | 80.7 |
> | CPO (COMET)         |  81.1 |  81.5 |  84.5 |  82.9 | 82.5 |
> | **Calibration (COMET)**  | **82.1** | **81.9** | **84.3** | **83.8** | **83.0** |
>
>
> It is easy to see that our calibration method consistently outperforms both SFT and CPO across all metrics, demonstrating its effectiveness in paragraph-level evaluation. That said, document-level evaluation remains an open challenge in current MT research. Practical measures are still under development and need to account for broader aspects such as consistency. These issues are beyond the scope of this paper and are left for future work.
>
> [1] WMT24++: Expanding the Language Coverage of WMT24 to 55 Languages & Dialects
>
> [2] Translating Step-by-Step: Decomposing the Translation Process for Improved Translation Quality of Long-Form Texts
>
> [3] Training and Meta-Evaluating Machine Translation Evaluation Metrics at the Paragraph Level
>
> > "l. 218ff. / l.333f.: I don't think it is justified to say that the quality is "comparable to that of the current top-performing system, that is Tower-70B-v2 equipped with 100-time-sampling MBR/TRR"as there are still significant gaps towards that from Table 1 which are as big (and sometimes even bigger) than the gain achieved through this approach compared to CPO."
>
> Let us clarify the comparison. In Table 1, our model, ​**TowerInstruct-Mistral-7B + Calibration**​, compared to ​**Tower-70B-v2 + MBR/TRR**​, achieves scores of 72.4 vs. 72.3 in en-de, 73.9 vs. 74.5 in en-es, 73.6 vs. 74.2 in en-ru, and 72.6 vs. 72.6 in en-zh. **On average, it slightly lags behind Tower-70B-v2 + MBR/TRR (73.1 vs. 73.4). However, our model outperforms CPO across these four directions by +1.0 (73.1 vs. 72.1)**, and by +1.0 across all directions (73.9 vs. 72.9).
>
> It is worth noting that Tower-70B+MBR/TRR is far from practical: (1) the base model is 10 times larger than our setting, and (2) MBR/TRR requires 100 times sampling followed by reranking. In contrast, our results rely solely on efficient MAP decoding, using beam search with a beam size of 5.
>
> ---
>
> Thank you again for your careful reading of our paper and for your insightful suggestions and questions. Hopefully, our response addresses your concerns. We will incorporate this discussion in our next version. Also, we are more than happy to provide further explanation or discussion if needed.

---

> > ### Comment · Reviewer_RNS9 · 2025-07-31
> > **Thank you for the rebuttal!**
> >
> > Great to see that PL works better but not as good as the Calibration method proposed here. The additional experiments really helped to strengthen my view on the paper and I will raise my score accordingly.

---

### Official Review · Reviewer_bA3p · 2025-06-25

**Clarity:** 3
**Significance:** 3
**Originality:** 3
**Rating:** 5
**Confidence:** 4

**Summary:**

The paper propose a method to calibrate the likelihood of translation hypotheses with their actual translation quality by optimizing the Pearson correlation between these two metrics during training, which aims to enhance the effectiveness of decoding and improve translation quality.
This calibration method involves sampling multiple hypotheses, computing their likelihoods and quality scores using an external metric (e.g., COMET), and minimizing the negative Pearson correlation between these scores.

The experiments conducted demonstrate significant improvements in translation quality across various metrics and languages when using this calibration method. Additionally, the calibrated likelihoods serve as a capable proxy for translation quality, surpassing some existing quality estimation models.

**Questions:**

Shown in weakness part, 1 for the SLiC paper comparison, 2 for the discrepancy between automatic evaluation improvement and human evaluation improvement.

**Ethical Concerns:**

["NO or VERY MINOR ethics concerns only"]

**Final Justification:**

My concern of the overlap with SLiC paper is properly addressed.
it's nice to see Calibration yields clear improvements over both SFT and PL.

**Limitations:**

yes

**Quality:**

3

**Strengths And Weaknesses:**

Strength:

First, I think the idea of applying likelihood calibration with quality estimation for machine translation task is novel, the method is simple and clear, which is also orthogonal to SFT method.

Second, give credit to the author for carrying out extensive experiments across 10 languages, multiple base models (Tower, ALMA), and metrics (COMET, XCOMET, KIWI) demonstrate consistent, significant gains over strong baselines. Human evaluations (200 samples/direction) confirm quality improvements.

Weakness:

I think the paper bears some similarity with the paper "Calibrating Sequence likelihood Improves Conditional Language Generation" in ICLR2023 (which the author has already cited.) which proposed a sequence likelihood calibration (SLiC) stage after MLE to enhance the model quality.  Although the metric and task may differ a little bit (this paper focus on MT while the SLiC paper doesn't cover ), I think the core idea is very relevant. So it's kind of bring down the innovativeness a bit.  So I would like to see more analysis on this part, may be a direction comparison in experiment.

Another small concern is that the translation quality improvement of human evaluation shown in Figure 4 is far more significant than the automatic evaluation improvement in Table1. Does it mean that the automatic evaluation metric like XCOMET and KIWI-XL could not fully reflect the improvement of this method ?  If so, is it problematic (or can be improved) to use COMET as the translation quality score for likelihood calibration.

---

> ### Author Rebuttal · Authors · 2025-07-31
>
> Thanks for your insightful suggestions and for noting our contribution. We summarize and address your concerns as follows:
>
> > The relevance and differences compared to the SLiC paper.
>
> Thank you for your careful reading and valuable suggestions. Our paper indeed shares part of the motivation with SLiC [Zhao et al., 2023][1]. We agree that a direct comparison would ideally strengthen our work. Unfortunately, [Zhao et al., 2023] did not release their codebase, and the implementation complexity, along with a lack of certain details, currently prevents us from conducting a direct comparison.
>
> Broadly speaking, SLiC can be viewed as a preliminary variant of PL-based preference optimization. Although indirect, it operates in a similar manner to [Zhu et al., 2024][2], a PL-based PO method applied in the machine translation domain. **To address your concern, we provide a direct comparison with this approach to demonstrate the effectiveness of our method.**
>
> To ensure comparability, (1) we follow their setting and apply our method on Mistral using their data (WMT data and human annotations). Also, we report our results when applying to the top-performing translation model, i.e., TowerMistral-7B [3], using their data. (2) We use the same beam search size of 4 in [Zhu et al., 2024]. The results are evaluated on the WMT22 test set using COMET as shown in the following table:
>
> |                            | EN-DE | DE-EN | EN-ZH | ZH-EN | Avg. |
> |----------------------------|:-----:|:-----:|:-----:|:-----:|:----:|
> | Mistral-Ins.               | 81.2  | 82.7  | 82.5  | 77.7  | 81.0 |
> | Mistral-Ins. + BEST        | 81.8  | 83.0  | 83.3  | 78.3  | 81.6 |
> | Mistral-Ins. + PL          | 82.9  | 83.4  | 84.7  | 79.3  | 82.6 |
> | **Mistral-Ins. + Calibration** | **83.5** | **84.4** | **85.4** | **80.1** | **83.4** |
> | TowerMistral + BEST        | 88.9  | 85.7  | 89.3  | 83.3  | 86.8 |
> | **TowerMistral + Calibration** | **89.9** | **86.6** | **90.1** | **84.7** | **87.8** |
>
> \+ Best is supervised fine-tuning using best scoring samples, and \+ PL is the result of their implementation of the PL model [2]. \+ Calibration is applying our method. It is easy to see that
>
> 1. On the weaker base models, such as Mistral-Ins, **Calibration yields clear improvements** over both SFT and PL.
> 2. Even when applying calibration to the top-performing system, TowerMistral, our system still achieves clear, albeit relatively smaller, gains, demonstrating that calibration training is also feasible with human annotations rather than machine-generated signals.
>
> Further, our work encompasses broader underlying concepts compared to [Zhao et al., 2023] in motivation, modeling, and analysis. We summarize and discuss them as follows:
>
> 1. [Zhao et al., 2023] aim to optimize candidate ranking based on their similarity to a target output, where similarity is computed using token or sentence representations. In contrast, our work proposes optimizing the Pearson correlation between translation likelihood and quality, rather than similarity. Moreover, the Pearson objective normalizes both likelihood and quality scores within a group, making it invariant to scale and shift. Our calibration approach directly incorporates translation quality via the metric function q(⋅∣x), optimizing at the distribution level using the Pearson correlation. By contrast, [Zhao et al., 2023] focus on pairwise comparisons, lacking a more direct utilization of absolute quality values at a distribution level.
> 2. In Section 5.2, we show that our objective, Pearson correlation itself, shares the goal with translation meta-evaluation. It provides a theoretical foundation for unifying translation quality optimization and quality evaluation within a single model.
> 3. In Section 5.3, we analyze the connection between calibration method and quality-aware decoding, and show that calibration captures the benefits of Best-of-N sampling (a key form of QAD) during training time. This, in turn, enhances the effectiveness of MAP decoding strategies such as beam search.
>
> [1] Calibrating Sequence Likelihood Improves Conditional Language Generation
>
> [2] A Preference-driven Paradigm for Enhanced Translation with Large Language Models
>
> [3] Tower: An Open Multilingual Large Language Model for Translation-Related Tasks
>
> > “Another small concern is that the translation quality improvement of human evaluation shown in Figure 4 is far more significant than the automatic evaluation improvement in Table 1. Does it mean that the automatic evaluation metrics like XCOMET and KIWI-XL could not fully reflect the improvement of this method? If so, is it problematic (or can be improved) to use COMET as the translation quality score for likelihood calibration.”
>
> Thanks for these insightful questions. We try to address them as follows:
>
> 1. Firstly, in Table 1, our method achieves +2.8 KIWI-XL and +2.7 XCOMET improvements. It is worth noting that these gains are **indeed big**. [Kocmi et al., 2024][4] provide a collective understanding of the meaning of differences in different metrics, emperically showing how big a metric difference is reflected in human annotator judgments. According to their finding, improvements of **≥ 1.99 in XCOMET or ≥ 0.94 in COMET** scores correspond to at least **90%** estimated agreement in human judgment—both of which are achieved by our method. To some extent, this aligns with our human annotation in EN-ZH and EN-RU, where the win rates are also high, e.g., 106 (win) / 32 (loss), and 93 (win) / 44 (loss).
> 2. Secondly, each metric has its own downside; COMET series scores are the same. The training data for COMET models comes from human annotations of each year's WMT data, which means they may be effective only within a limited range of translation quality, as WMT submissions are generally not bad. For example, [Lo et al., 2023][5] show that the COMET series struggles to evaluate translations when quality falls below a certain level. Accordingly, we are not sure, but it might also lack of ability to measure a system that outperforms previous years' systems, e.g., our method's output. We are not sure whether it fully reflects our system improvement. That is also a motivation for us to conduct human annotation.
> 3. Broadly speaking, although not perfect, using COMET-like neural metrics as quality signals for calibration is a successful practice, at least for mid- and high-resource languages that are well supported by current neural models. A potential improvement could be to combine multiple metrics to better exploit their complementary strengths and mitigate individual biases. Incorporating recent advanced metrics, such as LLM-based MQM, may also offer additional benefits. That said, we acknowledge that our method may not generalize well to certain low-resource languages, such as Bhojpuri, where neural metric support is lacking. Addressing these challenges will require collaborative efforts from the broader community, particularly in advancing evaluation metrics, which lie beyond the scope of this paper.
>
> [4] Navigating the Metrics Maze: Reconciling Score Magnitudes and Accuracies
>
> [5] Metric Score Landscape Challenge (MSLC23): Understanding Metrics’ Performance on a Wider Landscape of Translation Quality
>
> ---
>
> Thank you again for your insightful suggestions on our paper. We hope that our additional experiments and explanations address your concerns. We will incorporate our discussion into the next version. Please don’t hesitate to let us know if further clarification or discussion is needed; we’d be happy to provide it.

---

> > ### Comment · Reviewer_bA3p · 2025-08-05
> >
> > Thanks for the feedback.
> > My concern of the overlap with SLiC paper is properly addressed.
> > it's nice to see Calibration yields clear improvements over both SFT and PL.
> > I will raise my score accordingly.

---

### Official Review · Reviewer_FndS · 2025-06-27

**Clarity:** 3
**Significance:** 3
**Originality:** 3
**Rating:** 4
**Confidence:** 4

**Summary:**

This paper introduces a method to enhance machine translation for LLMs by calibrating the model's hypothesis likelihood with translation quality. At small computational cost, the method substantially improves translation decoding quality. Experiments on two top-performing translation-specialized LLMs (ALMA and Tower, 7B/13B parameters) demonstrate its efficacy. Furthermore, analysis experiments indicate that the model also improves its translation assessment capability.

**Questions:**

1. The comparison between the MBR of the Tower-70B model and the additionally trained 7B model are not univariate. It is suggested to supplement with an SFT fine-tuned or fine-tuned with proposed method on 7B model to perform MBR, to better control variables and demonstrate the effect of the proposed method when superimposed with MBR.
2. Supplement experiments or extra explanations for the points mentioned in the weakness part.

**Ethical Concerns:**

["NO or VERY MINOR ethics concerns only"]

**Final Justification:**

Thank you for your reply. Considering your reply and other reviewers' comments, I tend to maintain my current positive score.

**Limitations:**

yes

**Quality:**

3

**Strengths And Weaknesses:**

Strengths：
1. This paper proposes a simple and effective method presented in clear writing style, which is easy to follow.
2. The method is computationally economical compared with methods like MBR, and even achieves a higher performance.
3. Thorough experiments demonstrate significant improvement at low computational cost, reflected in both COMET-based metrics and human evaluation.
4. Analysis experiments indicate enhanced translation quality assessment ability, further illustrating the effectiveness of translation quality data calibration.

Weaknesses：
1. The experimental baselines are insufficient, only comparing the single baseline of CPO. Fine-tuning work based on translation quality preferences, such as the work [1] referenced by Section 2, should be considered for inclusion in the main experiments.
2. The experimental setup mentions that the presented results show the best outcomes under different learning rate settings. Sensitivity analysis regarding learning rates should be added to provide more information about the method’s robustness.

[1] A Preference-driven Paradigm for Enhanced Translation with Large Language Models

---

> ### Author Rebuttal · Authors · 2025-07-31
>
> Thanks for your insightful suggestions on our paper and highlighting its contributions. We address your concerns as follows:
>
> > "The experimental baselines are insufficient, only comparing the single baseline of CPO. Fine-tuning work based on translation quality preferences, such as the work [1] referenced by Section 2, should be considered for inclusion in the main experiments."
>
> Thank you for your suggestion. We agree that a direct comparison would ideally strengthen our work. Unfortunately, [Zhu et al., 2024] did not release their codebase, and their implementation is based on weaker models such as LLaMA-1 or Mistral. To address your concern, we apply our method on Mistral using their data (WMT data and human annotations). Also, we report our results when applying calibration to the top-performing translation model, i.e., TowerMistral-7B [1], using their data.
>
> To ensure comparability, we use the same beam search size of 4. The results are evaluated on the WMT22 test set using COMET as shown in the following table:
>
> |                            | EN-DE | DE-EN | EN-ZH | ZH-EN | Avg. |
> |----------------------------|:-----:|:-----:|:-----:|:-----:|:----:|
> | Mistral-Ins.               | 81.2  | 82.7  | 82.5  | 77.7  | 81.0 |
> | Mistral-Ins. + BEST        | 81.8  | 83.0  | 83.3  | 78.3  | 81.6 |
> | Mistral-Ins. + PL          | 82.9  | 83.4  | 84.7  | 79.3  | 82.6 |
> | **Mistral-Ins. + Calibration** | **83.5** | **84.4** | **85.4** | **80.1** | **83.4** |
> | TowerMistral + BEST        | 88.9  | 85.7  | 89.3  | 83.3  | 86.8 |
> | **TowerMistral + Calibration** | **89.9** | **86.6** | **90.1** | **84.7** | **87.8** |
>
> \+ Best is supervised fine-tuning using best scoring samples, and \+ PL is the result of their implementation of the PL model [2]. \+ Calibration is applying our method. It is easy to see that
>
> 1. On the weaker base models, such as Mistral-Ins, **Calibration yields clear improvements** over both SFT and PL. We **provisionally attribute** this gain to Calibration’s broader consideration of factors compared to PL. Specifically: (1) Calibration normalizes both likelihood and quality scores within a group, making the objective invariant to scale and shift. This promotes stable and robust optimization across diverse input distributions. (2) Calibration directly integrates translation quality via the metric function q(⋅∣x), and optimizes at the distribution level using the Pearson correlation. In contrast, [2] introduces reweighting for each pairwise comparison. In extreme cases, this approach may treat a comparison between ratings 2 (bad) and 1 (nonsense) the same as one between 4 (nearly perfect) and 3 (good), which could be overly heuristic.
> 2. Even when applying calibration to the top-performing system, TowerMistral, using human-annotated data, our system still achieves clear, albeit relatively smaller, gains, demonstrating that calibration training is also feasible with human annotations rather than machine-generated signals.
>
> We will include these analyses in our next version.
>
> [1] Tower: An Open Multilingual Large Language Model for Translation-Related Tasks
> [2] A Preference-driven Paradigm for Enhanced Translation with Large Language Models
>
> > "The experimental setup mentions that the presented results show the best outcomes under different learning rate settings. Sensitivity analysis regarding learning rates should be added to provide more information about the method’s robustness."
>
> Thank you for your careful reading. Yes, as mentioned in lines 195–196, to ensure robust results, we experiment with learning rates ranging from 1e-5 to 1e-4, in steps of 1e-5, for each setting.
>
> Due to space limitations, we present a subset of these results here. The following table (aligned with experiments in Table-1) provides the details, using TowerMistral-7B as the base model and CometKIWI-XL as the evaluation metric.
>
> |                    | en-de | en-es | en-ru | en-zh | en-fr | en-nl | en-it | en-pt | en-ko | Avg. |
> |--------------------|-------|-------|-------|-------|-------|-------|-------|-------|-------|------|
> | SFT (1e-5)         |  70.7 |  71.6 |  70.6 |  70.3 |  68.5 |  72.2 |  71.6 |  72.8 |  76.3 | 71.6 |
> | SFT (3e-5)         |  70.8 |  71.7 |  70.7 |  70.4 |  68.4 |  72.3 |  71.6 |  72.6 |  76.2 | 71.6 |
> | **SFT (5e-5)**     |  **70.7** |  **71.8** |  **70.8** |  **70.5** |  **68.5** |  **72.3** |  **71.6** |  **72.7** |  **76.2** | **71.7** |
> | SFT (7e-5)         |  70.8 |  71.8 |  71.1 |  70.6 |  68.5 |  72.3 |  71.8 |  72.7 |  76.1 | 71.7 |
> | SFT (9e-5)         |  70.9 |  71.9 |  70.9 |  70.6 |  68.4 |  72.2 |  71.7 |  72.7 |  76.2 | 71.7 |
> | CPO (1e-5)         |  71.3 | 72.7 | 72.5 | 71.8 | 69.6 | 72.9 | 72.8 | 73.7 | 77.2 | 72.7 |
> | CPO (3e-5)         |  71.3 |  72.7 |  72.6 |  72.0 |  69.6 |  73.2 |  73.2 |  74.0 |  77.2 | 72.9 |
> | **CPO (5e-5)**     |  **71.2** |  **73.1** |  **72.3** |  **71.8** |  **70.0** |  **73.3** |  **73.1** |  **74.0** |  **77.4** | **72.9** |
> | CPO (7e-5)         |  71.2 |  73.0 |  72.7 |  71.9 |  69.6 |  73.1 |  72.9 |  74.1 |  77.1 | 72.8 |
> | CPO (9e-5)         |  71.1 |  72.8 |  72.4 |  71.6 |  69.5 |  72.8 |  72.6 |  73.7 |  76.9 | 72.6 |
> | Calibration (1e-5) |  72.3 |  73.9 |  73.2 |  72.4 |  70.5 |  74.0 |  73.9 |  74.7 |  78.1 | 73.7 |
> | Calibration (3e-5) |  72.3 |  73.6 |  73.5 |  72.6 |  70.4 |  74.1 |  73.8 |  74.8 |  78.0 | 73.7 |
> | **Calibration (5e-5)** |  **72.4** |  **73.9** |  **73.6** |  **72.6** |  **70.8** |  **74.2** |  **74.1** |  **75.1** |  **78.1** | **73.9** |
> | Calibration (7e-5) |  72.4 |  73.9 |  73.4 |  72.7 |  70.5 |  74.1 |  73.7 |  74.8 |  78.1 | 73.7 |
> | Calibration (9e-5) |  72.1 |  73.6 |  73.3 |  72.5 |  70.4 |  73.4 |  73.4 |  74.7 |  77.0 | 73.4 |
>
> The average results for all 10 steps are as follows.
>
> | Learning Rate              | 1e-05 | 2e-05 | 3e-05 | 4e-05 | 5e-05 | 6e-05 | 7e-05 | 8e-05 | 9e-05 | 1e-04 |
> |----------------------------|---------:|---------:|---------:|---------:|---------:|---------:|---------:|---------:|---------:|---------:|
> | SFT (CometKIWI-XL)         |     71.6 |     71.6 |     71.6 |     71.6 |     **71.7** |     71.7 |     71.7 |     71.7 |     71.7 |     71.7 |
> | CPO (CometKIWI-XL)         |     72.7 |     72.8 |     72.9 |     72.9 |     **72.9** |     72.3 |     72.8 |     72.8 |     72.6 |     72.6 |
> | Calibration (CometKIWI-XL) |     73.7 |     73.6 |     73.7 |     73.6 |     **73.9** |     73.6 |     73.7 |     73.5 |     73.4 |     73.5 |
>
> During our experiments, we found that SFT is not particularly sensitive within the explored learning rate range, which we attribute to the extensive high-quality SFT already performed on TowerMistral-7B.
> For CPO and Calibration, the performance tends to fluctuate, often degrading at relatively small or large learning rates. The optimal learning rate for both is generally around 5e-5 across different evaluation settings.
>
> > "The comparison between the MBR of the Tower-70B model and the additionally trained 7B model is not univariate. It is suggested to supplement with an SFT fine-tuned or fine-tuned with the proposed method on 7B model to perform MBR, to better control variables and demonstrate the effect of the proposed method when superimposed with MBR."
>
> Thank you for your suggestion. Indeed, directly conducting experiments on the base model with MBR applied would better demonstrate the effect of our method. However, applying the same magnitude of MBR (e.g., 100-fold sampling) to TowerMistral is challenging due to our limited computational resources. (That said, we are currently running such an experiment based on your suggestion, and we hope to provide updated results before the rebuttal period ends.)
>
> Although indirect, in Section 5.3, we provide a comparison of calibration with BoN (sampling and reranking using QE model), another key form of quality-aware decoding that is similar to MBR. We summarize them as follows:
>
> In Section 5.3, we present direct results under settings of (1) base-model + beam-search, (2) base-model + BoN, (3) calibration-model + beam-search, and (4) calibration-model + BoN. To ensure fairness, we adopt a cross-metric evaluation: Candidates are reranked using KIWI-XXL, while the results are measured using a reference-based metric, Comet22:
>
> |       Settings                    | Comet22 |
> |---------------------------|---------|
> | Tower-Base (Beam=5)       | 82.6    |
> | Tower-Base (BoN=10)       | 82.8    |
> | Tower-Base (BoN=100)      | 83.4    |
> | Tower-Calibrate (Beam=5)  | 83.9    |
> | Tower-Calibrate (BoN=10)  | 83.3    |
> | Tower-Calibrate (BoN=100) | 84.0    |
>
> In short, we can easily see that:
>
> 1. Our calibrated model with beam search (beam=5) achieves clearly better performance than base model + BoN (100-time sampling);
> 2. When controlling the model as Tower-Base,  beam search (beam=5) leg behind that of BoN, even with only 10-time sampling;
> 3. When controlling the model as our calibrated model,  beam search (beam=5) achieves comparable results with that of BoN (100-time sampling).
>
> **These results form the basis of our conclusion in Section 5.3: calibration improves the effectiveness of maximum *a posteriori* decoding and offers clear efficiency advantages over QAD.**
>
> ---
>
> Thank you again for your careful reading and meaningful suggestions. We hope our additional explanation and experiments address your concern. We look forward to your further discussion!

---

> > ### Comment · Reviewer_FndS · 2025-08-04
> > **Thank you for your response**
> >
> > Thank you for your response, and I will keep the score which is learning positive for the acceptance.

---

### Official Review · Reviewer_jW1Y · 2025-07-03

**Clarity:** 3
**Significance:** 3
**Originality:** 3
**Rating:** 5
**Confidence:** 4

**Summary:**

This paper addressed a well-known gap in translation decoding, where employing maximum a posteriori (MAP) search often generates lower-quality outputs, as model probability doesn't align well with human judgements. The issue is known as miscalibration issue. To resolve it, the authors proposed calibrating translation likelihoods with quality estimates, specifically, they train a lightweight calibration layer on top of LLMs, optimizing the Pearson correlation between predicted likelihood and reference quality. With just 2k instances per language direction, this quality-calibrated likelihood consistently enhances translation quality that are detected both in automatic evaluation metrics and human evaluations. Tested across 10 languages and on translation-specialized LLMs like TowerMistral, the authors confirmed the consistent gains by using the proposed approach.

- the paper is mostly easy to follow.
- looks like 2k samples suffice, but I am curious how smaller or larger sample sizes impact performance. If possible, additional experiments in mid/low-resourced directions would be interesting if the page allows.
- do you have any insights into search dynamics of the proposed method? Compared to MAP decoding, does calibration impact beam search diversity or risk repetitive patterns or hallucination unnecessarily?

**Questions:**

please see the summary.

**Ethical Concerns:**

["NO or VERY MINOR ethics concerns only"]

**Final Justification:**

In response from the authors, my concern and questions are well addressed. I raise my score 4->5.

**Quality:**

3

**Strengths And Weaknesses:**

strengths
- the proposed method works well with minimal data of 2k examples per direction and computational resources of 1-2 H100 GPUs, making it practical and efficient.
- Experimental results across multiple languages, metrics, and human evaluations demonstrate improvements
- the authors will likely make available code and model checkpoints, enabling reproducibity

weaknesses
- unclear to me how well the calibration holds under domain shifts or when scaling to low-resource languages with low-quality noisier signals. It'd be interesting to report the results for mid/low-resourced language directions.
- this work would benefit from deeper comparisons to related methods like quality-aware decoding [Fernandes et al., 2022]. Have you ever done such experiments?

---

> ### Author Rebuttal · Authors · 2025-07-30
>
> Thank you for carefully reading our paper and highlighting its contributions. We address your concerns as follows:
>
> > "Unclear to me how well the calibration holds under domain shifts or when scaling to low-resource languages with low-quality, noisier signals. It'd be interesting to report the results for mid/low-resourced language directions."
>
> In fact, our experiments with ALMA already include a low-resource language: Icelandic. As shown in Tables 4 and 5, ALMA achieves notable improvements for **EN→IS (+0.9 COMET, +2.6 COMETKIWI, +1.7 XCOMET)** and **IS→EN (+1.2 COMET, +1.7 COMETKIWI, +4.0 XCOMET)**, consistently outperforming both SFT and CPO baselines.
>
> To further demonstrate the effectiveness of calibration in low-resource settings, we conducted one additional experiment. Since Tower and ALMA primarily target high-resource languages, we selected **Gemma-3-12B** (google/gemma-3-12b-it) as our testbed, due to its strong multilingual base capabilities and broader language coverage.
>
> Settings: (1) We applied our calibration method directly to Gemma-3-12B across 10 languages directions from this year’s WMT25 competition (ongoing), covering low-resource languages (Icelandic, Estonian) and mid-resource languages (Serbian, Ukrainian), as well as six other 6 high-resource languages. (2) We used the same data generation pipeline described in our paper, originally based on ​Flores; (3) We reused the same training hyperparameters as those used for ​TowerMistral-7B. The results are as follows:
>
> |             | Metric  | en-zh | en-ar | en-cs | en-ja | en-ko | en-ru | en-sr (mid) | en-uk (mid) | en-et (low) | en-is (low) | Avg. |
> |-------------|---------|:-----:|:-----:|:-----:|:-----:|:-----:|:-----:|:-----------:|:-----------:|:-----------:|:-----------:|:----:|
> | SFT         | KIWI-XL |  70.8 |  65.7 |  68.9 |  75.1 |  76.0 |  71.4 |     71.3    |     69.3    |     71.9    |     66.8    | 70.7 |
> | CPO         | KIWI-XL |  72.1 |  67.0 |  70.2 |  76.5 |  77.4 |  72.4 |     71.4    |     70.3    |     73.4    |     68.7    | 71.9 |
> | **Calibration** | KIWI-XL |  **72.9** |  **67.9** |  **71.4** |  **77.1** |  **77.9** |  **73.6** |     **72.9**    |     **71.4**    |     **75.1**    |     **69.2**    | **72.9** |
> | SFT         | Comet22   |  80.2 |  73.4 |  81.3 |  83.9 |  83.8 |  80.8 |     82.6    |     80.4    |     82.1    |     75.4    | 80.4 |
> | CPO         | Comet22   |  80.7 |  74.1 |  81.3 |  84.3 |  84.1 |  80.9 |     82.3    |     80.4    |     82.4    |     76.1    | 80.7 |
> | **Calibration** | Comet22   |  **80.9** |  **74.7** |  **82.1** |  **84.6** |  **84.4** |  **81.5** |     **83.2**    |     **81.0**    |     **83.0**    |     **76.6**    | **81.2** |
>
> We can see that in this setting, for mid- and low-resource languages (the last four columns), calibration yields clear improvements over both SFT and CPO across both reference-free and reference-based metrics, aligning with our paper. That said, we acknowledge that for certain low-resource languages lacking support from neural metric models, such as Bhojpuri, our method is likely to fail. Addressing these directions will require joint efforts from the broader community, particularly in developing better evaluation metrics, which fall beyond the scope of this paper.
>
> > "This work would benefit from deeper comparisons to related methods like quality-aware decoding [Fernandes et al., 2022]. Have you ever done such experiments?
>
> **Yes, we do have such experiments**. Quality-aware decoding (QAD), particularly in the context of [Fernandes et al., 2022], typically involves reranking or voting strategies. A common instantiation of QAD is Best-of-N (BoN) sampling, where translations are scored and ranked using a reference-free metric model (e.g., COMETKiwi-XXL). In Section 5.3, we present direct results under settings of (1) base-model + beam-search, (2) base-model + BoN, (3) calibration-model + beam-search, and (4) calibration-model + BoN.
>
> To ensure fairness, we adopt a cross-metric evaluation: Candidates are reranked using KIWI-XXL, while the results are measured using a reference-based model, Comet22. To be fair, for calibration, we also use KIWI-XXL as the training signal. We summarize them as follows:
>
> |       Settings                    | Comet22 |
> |---------------------------|---------|
> | Tower-Base (Beam=5)       | 82.6    |
> | Tower-Base (BoN=10)       | 82.8    |
> | Tower-Base (BoN=100)      | 83.4    |
> | Tower-Calibrate (Beam=5)  | 83.9    |
> | Tower-Calibrate (BoN=10)  | 83.3    |
> | Tower-Calibrate (BoN=100) | 84.0    |
>
> In short, we can easily see that:
>
> 1. Our calibrated model with beam search (beam=5) achieves clearly better performance than base model + BoN (100-time sampling);
> 2. When controlling the model as Tower-Base,  beam search (beam=5) leg behind that of BoN, even with only 10-time sampling;
> 3. When controlling the model as our calibrated model,  beam search (beam=5) achieves comparable results with that of BoN (100-time sampling).
>
> **These results form the basis of our conclusion in Section 5.3: ​Calibration enhances the effectiveness of maximum *a posteriori* decoding. Also, compared to QAD, it offers clear efficiency benefits.** Note that 100-time sampling is far from practical for online services. From another perspective, calibration can be seen as a training time optimization that captures the benefits of QAD.
>
> > "Do you have any insights into search dynamics of the proposed method? Compared to MAP decoding, does calibration impact beam search diversity or risk repetitive patterns or hallucination unnecessarily?"
>
> Thanks for your insightful questions! In our paper, we did not dig deep into the search dynamics or changes in the decoding space. But we can share with you some observations during our experiments.
>
> 1. **Hallucination**: We did not observe any hallucinations (lexicon/pattern repetition) during our experiments, at least not in Chinese, German, Dutch, and Russian, where we conducted detailed human analysis. In fact, even the base model (TowerMistral-7B) exhibited only minimal instances of hallucination. While not rigorously, we have a feeling that hallucinations are largely due to limited instruction-following ability, which can be effectively mitigated through SFT with a few translation examples, at least for high- and mid-resource languages. Notably, in our additional experiments with the instruction-tuned Gemma-12B model, we also found no hallucination issues, even without SFT stage.
> 2. **Beam Search Diversity**: Broadly speaking, calibration is to reorder the hypothesis likelihood within decoding space, while its impact on the shape of the space (flatter or sharper) is hard to say. We believe it depends on multiple aspects, like the sampling strategy.
>    Additionally, we read your question about diversity as a concern that calibration might reduce the diversity of the decoding space, potentially diminishing the benefits of using a larger beam size. To investigate this, we conducted an experiment using the TowerMistral-7B model, selecting four language directions (EN-DE, EN-ES, EN-RU, EN-ZH) and varying the beam size to compare its impact on both the base and our calibrated model as follows. It is easy to see that: (1) When the beam size exceeds 5, we do not observe further gains. (2) From Beam-1 to Beam-5, the gains for the SFT model and Calibrated models are comparable, 0.7 vs. 0.8 in KIWI-XL and 0.4 vs. 0.6 in Comet22. **Thus, we can assume that calibration does NOT change the shape of the space to a degree that affects search effectiveness.**
>
> | Beam-Size             |  1 |  2 |  3 |  4 |  5 |  6 |  7 |  8 |  9 | 10 |
> |-----------------------|:----:|:----:|:----:|:----:|:----:|:----:|:----:|:----:|:----:|:----:|
> | SFT (KIWI-XL)         | **70.3** | 70.8 | 70.9 | 70.9 | **71.0** | 70.9 | 70.9 | 70.9 | 70.9 | 70.9 |
> | Calibration (KIWI-XL) | **72.3** | 72.8 | 73.1 | 73.1 | **73.1** | 73.2 | 73.2 | 73.2 | 73.2 | 73.2 |
> |                       |      |      |      |      |      |      |      |      |      |      |
> | SFT (Comet22)           | **82.2** | 82.4 | 82.6 | 82.6 | **82.6** | 82.6 | 82.6 | 82.6 | 82.6 | 82.6 |
> | Calibration (Comet22)   | **83.3** | 83.6 | 83.7 | 83.9 | **83.9** | 83.9 | 83.8 | 83.9 | 83.9 | 83.9 |
>
> ---
>
> Thank you again for your careful reading of our paper and for your insightful suggestions and questions. Hopefully, our response addresses your concerns. We will incorporate this discussion, particularly for the low-resource and decoding space aspects, in our next version. Also, we are more than happy to provide further explanation or discussion if needed.

---

> > ### Author Response · Authors · 2025-08-02
> > **Additional Response by Authors**
> >
> > We provide additional experiments regarding your questions about 1) the impact of data size and 2) beam search diversity.
> >
> > **The impact of data size**: For setting in Table 1, we randomly sample 100, 200, 400, 800, and 1600 data instances and conduct experiments to show the impact of data size. The following table shows the average performance across all language directions:
> >
> > | Sample Size |    0 |  100 |  200 |  400 |  800 | 1600 | All (2000) |
> > |-------------|-----:|:----:|:----:|:----:|:----:|:----:|:----------:|
> > | KIWI-XL     | 71.1 | 72.7 | 73.0 | 73.2 | 73.8 | 73.9 |    73.9    |
> > | XCOMET      | 86.7 | 88.1 | 88.5 | 88.8 | 89.2 | 89.4 |    89.4    |
> >
> > It is interesting that even for 100 samples, the gain is also clear. As the sample size exceeds 800, we can observe only limited further improvements. Due to time constraints, we did not scale the sample size further.
> >
> > **Beam search diversity**: We conduct a more targeted experiment to address your question about beam search diversity. Settings: (1) We collect the Top-5 outputs from beam search under both calibration and SFT, using the same setup and test data as in Table 1. (2) We compute the 2-gram diversity (at the subword level) across the Top-5 outputs, defined as the proportion of distinct 2-grams out of all 2-grams present in the five outputs.
> >
> > Thus, the diversity is defined as **len(set(output1_list + output2_list + ... + output5_list)) / len(output1_list + output2_list + ... + output5_list)**. The following table shows the results:
> >
> > |             | en-de | en-es | en-ru | en-zh | en-fr | en-nl | en-it | en-pt | en-ko |
> > |-------------|:-----:|:-----:|:-----:|:-----:|:-----:|:-----:|:-----:|:-----:|:-----:|
> > | SFT         | 0.277 | 0.271 | 0.272 | 0.276 | 0.274 | 0.270 | 0.225 | 0.273 | 0.259 |
> > | Calibration | 0.276 | 0.271 | 0.273 | 0.270 | 0.272 | 0.268 | 0.224 | 0.270 | 0.260 |
> >
> > Note that here the minimum diversity rate is 0.2 when all five outputs are identical (a theoretically impossible scenario), and the maximum diversity is 1.0 when no two outputs share any 2-gram tokens.
> >
> > We observe that: (1) beam search typically produces low output diversity, consistent with findings in MT research [1]; and **(2) the 2-gram diversity ratios for SFT and Calibration are very similar across all language directions, indicating that calibration did not impact beam search diversity a lot**.
> >
> > [1] A Systematic Exploration of Diversity in Machine Translation
> >
> > Thank you again for your valuable suggestions. We hope our additional experiments sufficiently address your questions and concerns. Also, we are happy to provide further explanation or discussion if needed.

---

> > > ### Comment · Reviewer_jW1Y · 2025-08-05
> > >
> > > Thank you for your response! I raise my score, accordingly.

---

### Note · Authors · 2025-08-14

We sincerely thank all reviewers for their careful reading and encouraging feedback. We are pleased that they found:

1. Our paper is clearly written and easy to follow [jW1Y, Fnds];
2. Our method is simple, efficient, and effective [jW1Y, FndS, bA3p];
3. The experiments are extensive and thorough [bA3p, FndS, RNS9]: Following the best practice and achieving SOTA performance, covering 10 languages, and evaluated via multiple metrics and human annotation;
4. Our method and findings are novel and interesting [bA3p, RNS9]: Optimizing translation performance and quality estimation within a single model, simultaneously.

We are pleased that all reviewers agree that our response addresses their concerns and grade positively in response:

1. Effectiveness for low-resource or long-form translation scenarios [jW1Y, RNS9];
2. Deeper comparison to quality-aware decoding [jW1Y, FndS];
3. Comparison with PL-based method [FndS, bA3p, RNS9].

We also thank all reviewers for their suggestions that make this paper better:
1. Exploring the search dynamics or changes in the decoding space [jW1Y];
2. Reporting sensitive analysis [FndS];
3. Providing better interpretation of experimental results [bA3p, RNS9].

We will incorporate these valuable insights into the next version to better reflect these discussions.

Sincerely yours,

Authors

---

### Decision · Program_Chairs · 2025-09-17

**Decision:**

Accept (poster)

**Comment:**

The paper proposes to calibrate NMT systems’ hypotheses likelihood to translation quality by directly optimizing the correlation between the two during training.This results in improved translation quality across many settings and language pairs for translation-specialized LLMs..

Reviewers note that the paper is easy to follow and clearly written (jW1Y, jW1Y, bA3p), the approach has good sample efficiency (jW1Y, FndS) and the experiments are comprehensive and show clear improvement in multiple settings (jW1Y, FndS, bA3p, RNS9). Reviewer RNS9 also note that the paper provides interesting and novel observations (calibration improves quality and QE, scale improves quality but not QE).

During rebuttal, the authors provided several additional comparisons in response to the discussion about the lack of baseline comparisons to prior works (SliC, PO) raised by reviewers (bA3p, RNS9, FndS), sensitivity to hyperparameters (FndS),  generalization of approach in new settings (low-resource LPs and paragraph-level data) raised by jW1Y, RNS9, which have significantly strengthen the  submission.

Given the methodological clarity, the empirical thoroughness (both in the original and updated results), and the relevance of the problem addressed, the consensus among reviewers lean toward a positive assessment, which I agree with.